# Integrative network analysis reveals molecular mechanisms of blood pressure regulation

Tianxiao Huan[1,2,†], Qingying Meng[3], Mohamed A Saleh[4,5], Allison E Norlander[4], Roby Joehanes[1,2,6,7,8,†], Jun Zhu[9,10], Brian H Chen[1,2,†], Bin Zhang[9,10], Andrew D Johnson[1,11,†], Saixia Ying[6,†], Paul Courchesne[1,2,†], Nalini Raghavachari[12,†], Richard Wang[13,†], Poching Liu[13,†], The International Consortium for Blood Pressure GWAS (ICBP), Christopher J O'Donnell[1,11,†], Ramachandran Vasan[1,†], Peter J Munson[6,†], Meena S Madhur[4], David G Harrison[4], Xia Yang[3,*] & Daniel Levy[1,2,†,**]

## Abstract

Genome-wide association studies (GWAS) have identified numerous loci associated with blood pressure (BP). The molecular mechanisms underlying BP regulation, however, remain unclear. We investigated BP-associated molecular mechanisms by integrating BP GWAS with whole blood mRNA expression profiles in 3,679 individuals, using network approaches. BP transcriptomic signatures at the single-gene and the coexpression network module levels were identified. Four coexpression modules were identified as potentially causal based on genetic inference because expression-related SNPs for their corresponding genes demonstrated enrichment for BP GWAS signals. Genes from the four modules were further projected onto predefined molecular interaction networks, revealing key drivers. Gene subnetworks entailing molecular interactions between key drivers and BP-related genes were uncovered. As proof-of-concept, we validated *SH2B3*, one of the top key drivers, using *Sh2b3*$^{-/-}$ mice. We found that a significant number of genes predicted to be regulated by *SH2B3* in gene networks are perturbed in *Sh2b3*$^{-/-}$ mice, which demonstrate an exaggerated pressor response to angiotensin II infusion. Our findings may help to identify novel targets for the prevention or treatment of hypertension.

**Keywords** blood pressure; coexpression network; gene expression; hypertension; systems biology

**Subject Categories** Genome-Scale & Integrative Biology; Molecular Biology of Disease
**Mol Syst Biol. (2015) 11: 799**

## Introduction

Blood pressure (BP) is a highly heritable physiological trait that is regulated through the interactions of numerous genes and environmental factors. Over one billion people worldwide suffer from hypertension (systolic BP [SBP] ≥ 140 mm Hg or diastolic BP [DBP] ≥ 90 mm Hg) (Kearney *et al*, 2005). BP elevation contributes to nearly half the deaths from cardiovascular disease (CVD) (Lawes *et al*, 2006; Ehret & Caulfield, 2013). BP control in hypertensive individuals, in turn, is an effective intervention for reducing CVD risk (Lewington *et al*, 2002). It is hoped that advances from understanding the molecular underpinnings of BP regulation will improve the prediction of CVD susceptibility and offer insights into personalized treatments for hypertension that can reduce the risk of its sequelae.

A recent genome-wide association study (GWAS) meta-analysis of up to 200,000 people identified 29 genetic variants (at 28 loci) associated with BP (Ehret *et al*, 2011). However, the proportion of interindividual BP variability explained by these genetic variants was only about 1% (Ehret *et al*, 2011). It has been increasingly

---

1   The National Heart Lung and Blood Institute's Framingham Heart Study, Framingham, MA, USA
2   The Population Sciences Branch and the Division of Intramural Research, National Heart, Lung and Blood Institute, Bethesda, MD, USA
3   Department of Integrative Biology and Physiology, University of California, Los Angeles, CA, USA
4   Department of Medicine, Division of Clinical Pharmacology, Vanderbilt University, Nashville, TN, USA
5   Department of Pharmacology and Toxicology, Faculty of Pharmacy, Mansoura University, Mansoura, Egypt
6   Mathematical and Statistical Computing Laboratory, Center for Information Technology, National Institutes of Health, Bethesda, MD, USA
7   Harvard Medical School, Boston, MA, USA
8   Hebrew SeniorLife, Boston, MA, USA
9   Institute of Genomics and Multiscale Biology, New York, NY, USA
10  Graduate School of Biological Sciences, Mount Sinai School of Medicine, New York, NY, USA
11  Cardiovascular Epidemiology and Human Genomics Branch, Division of Intramural Research, National Heart, Lung and Blood Institute, Bethesda, MD, USA
12  Division of Geriatrics and Clinical Gerontology, National Institute on Aging, Bethesda, MD, USA
13  Genomics Core facility Genetics & Developmental Biology Center, The National Heart, Lung and Blood Institute, Bethesda, MD, USA
    *Corresponding author. Tel: +1 310 206 1812; Fax: +1 310 206 9184; E-mail: xyang123@ucla.edu
    **Corresponding author. Tel: +1 508 935 3458; Fax: +1 508 872 2678; E-mail: levyd@nih.gov
    †This article has been contributed to by US Government employees and their work is in the public domain in the USA

recognized that genes, instead of working in isolation, interact with other genes in complex regulatory networks, such as gene coexpression networks comprised of modules of genes demonstrating high levels of coregulation (Zhang & Horvath, 2005; Langfelder & Horvath, 2008). Genetic variants associated with diseases can perturb specific parts of gene networks, termed subnetworks, whose overall dysregulation shifts homeostatic processes and leads to disease (Huan *et al*, 2013; Civelek & Lusis, 2014; Mäkinen *et al*, 2014). Along the same line, we hypothesized that the top genetic loci identified in BP GWAS, together with a large number of additional genetic variants with more subtle effects, drive shifts in specific gene subnetworks that in turn affect BP.

We designed a systems biology framework to integrate gene expression (in this study, gene expression refers to mRNA expression) profiles with BP GWAS and cellular network models as a means to explore molecular mechanisms influencing BP regulation (Fig 1). We conducted our research using genetic and transcriptomic data (17,318 measured gene transcripts) from 3,679 Framingham Heart Study (FHS) participants who were not receiving antihypertensive drug treatment. At first, we investigated the association of BP with transcriptomic changes at the individual-gene level by identifying differentially expressed genes for BP transcriptome-wide (i.e., the top BP signature gene set) and at the multiple-gene level by identifying BP-associated coexpression network modules (coEMs). To identify BP coEMs, we first constructed a coexpression network from the gene expression data from all 3,679 samples in order to capture coexpression modules containing highly coregulated genes across all individuals. We then identified BP coEMs whose eigengenes (representing the expression patterns of all genes in each module) demonstrated significant correlations with BP measurements. The advantage of a coexpression network approach is that it provides a contextual framework to determine the relationship between the phenotype and functionally related genes across a population. Second, to differentiate BP-correlated gene sets that are potentially causal, we linked the top BP signature gene set and the BP coEMs (both representing BP gene sets) with expression-associated single nucleotide polymorphisms (eSNPs) and with BP GWAS SNPs (Ehret *et al*, 2011) to identify genetically inferred causal BP gene sets. A BP gene set was considered causal by genetic inference if it was significantly enriched with eSNPs that demonstrated low *P*-values in BP GWAS (Ehret *et al*, 2011). Third, further integration of the observed genetically inferred causal BP gene sets with molecular networks including protein–protein interaction (PPI) networks (Keshava Prasad *et al*, 2009) and blood Bayesian networks (Emilsson *et al*, 2008) uncovered key driver (KD) genes that serve as network hubs by interconnecting genes in the genetically inferred causal BP gene sets. Forth, KDs were further ranked by leveraging their associations with BP in GWAS and their differential expression in relation to BP. Lastly, as proof-of-concept, we investigated the role of one of the top KDs, *SH2B3*, in relation to hypertension using a *Sh2b3*$^{-/-}$ mouse model, and tested the genes in the predicted SH2B3 subnetworks for enrichment with differentially expressed genes in the knockout mouse model.

## Results

### Clinical characteristics of study participants

The FHS recently launched the Systems Approach to Biomarker Research in Cardiovascular Disease (SABRe CVD) initiative, which seeks to explore and characterize biomarkers and molecular underpinnings of CVD and its risk factors, including BP. High-throughput gene expression profiles from whole-blood-derived RNA were generated in 5,626 individuals of European ancestry from the FHS offspring ($n = 2,446$) and the third-generation ($n = 3,180$) cohorts. In order to avoid the confounding effects of drug treatment on gene expression levels, this study was restricted to 3,679 participants who were not receiving antihypertensive treatment.

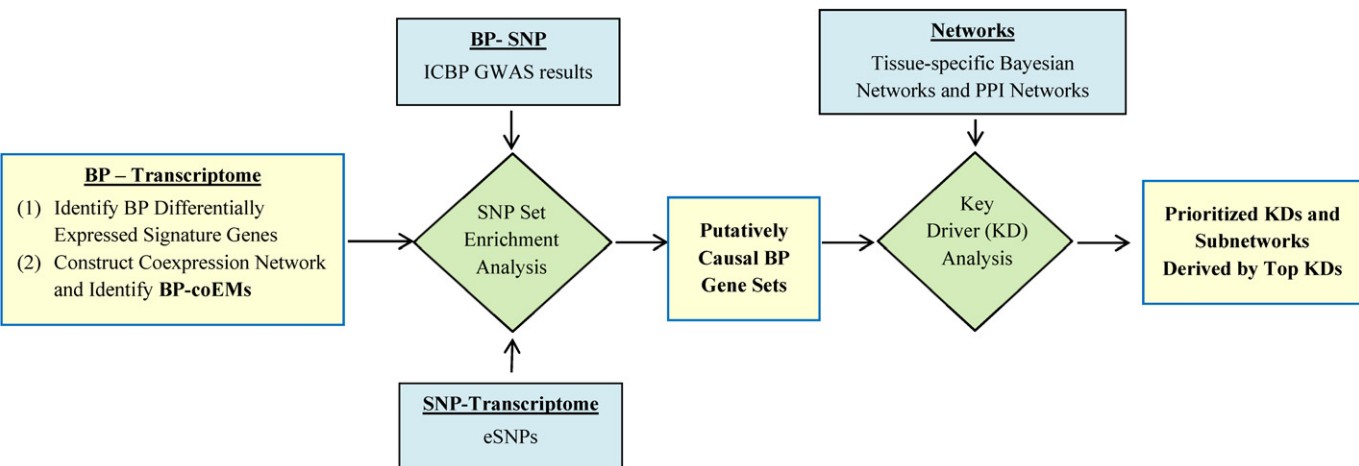

**Figure 1.  The integrative network-based approach for identifying and prioritizing key drivers of blood pressure regulation.**
This figure depicts the analysis framework. First, identify blood pressure (BP)-associated transcriptomic changes in individual-gene level by identifying differentially expressed signature genes, and in multiple-gene level by identifying BP-associated coexpression network modules (BP coEMs). Second, integrate both the BP top signatures gene set and the BP coEMs, with BP genome-wide association studies (GWAS) results as well as eSNPs by the SNP set enrichment method (SSEA) (Zhong *et al*, 2010) to identify genetically inferred causal BP gene sets. Third, project the genes of the genetically inferred causal BP gene sets onto network models to prioritize and identify key driver (KD) genes. Finally, identify BP-associated subnetworks derived by top KDs. ICBP = International Consortium for Blood Pressure, PPI = protein–protein interaction.

**Table 1. Clinical characteristics of FHS participants.**

| Phenotypes/Covariates | Offspring cohort N = 1,102 (examination cycle 8: 2005–2008)[a] Mean $\pm$ SD | Third-generation cohort N = 2,577 (examination cycle 2: 2008–2011)[a] Mean $\pm$ SD |
|---|---|---|
| Male (%) | 38 | 44 |
| Age (years) | 63 $\pm$ 9 | 45 $\pm$ 8 |
| Body mass index (kg/m²) | 27.1 $\pm$ 5.0 | 27.2 $\pm$ 5.4 |
| Systolic BP (mm Hg) | 126 $\pm$ 16 | 115 $\pm$ 14 |
| Diastolic BP (mm Hg) | 75 $\pm$ 10 | 74 $\pm$ 9 |
| Hypertension (%) | 21 | 7 |

[a]Individuals who were receiving antihypertensive treatment were excluded in this study.

The clinical characteristics of the 3,679 study participants are summarized in Table 1. The mean age of study samples was 51 years (range 24–92) and 58% were female. The distribution of systolic (SBP) and diastolic BP (DBP) is shown in Supplementary Fig S1. The average SBP/DBP was 118/74 mm Hg, and 11% of participants had hypertension (HTN; defined as SBP $\geq$ 140 mm Hg or DBP $\geq$ 90 mm Hg). Pre-hypertension, defined as a SBP from 120 to 139 mm Hg or DBP from 80 to 89 mm Hg, was present in 17% of our participants.

## Influence of blood cell types on BP-associated gene expression differences

As mRNA expression levels might be influenced by differences in the proportions of different cell types in whole blood, we assessed the correlations between mRNAs and three major cell-type proportions. We found that approximately 42% of genes were significantly correlated with cell-type proportions at Bonferroni-corrected $P < 0.05$ (Supplementary Table S1), suggesting a major impact of blood cell types on gene expression. As results from both cell type-adjusted and cell type-unadjusted analyses could be biologically relevant (the adjusted analysis may reflect cell-type-independent signals and the unadjusted analysis may represent cell-type-dependent signals), we report both sets of results but focus our discussions on the adjusted analysis to simplify results interpretation. We also report the similarities and differences between the two analyses.

## Identification of transcriptome-wide gene expression signatures for BP

To characterize significant transcriptomic changes at the single-gene level, we correlated each gene with BP phenotypes (SBP, DBP, and HTN) after accounting for age, sex, body mass index (BMI), cell types, technical covariates, and familial relatedness (detailed in the Materials and Methods section). Eighty-three genes whose expression levels were correlated with BP (73 for SBP, 31 for DBP, and eight for HTN) were identified at Bonferroni-corrected $P < 0.05$ (corrected for 17,318 measured genes) (Supplementary Dataset S1). These 83 signature genes are referred to as the BP signature gene set. Among the 83 genes, 65 were positively correlated with BP traits

and eight were negatively correlated. Five genes, *TSPAN2*, *GZMB*, *MYADM*, *ANXA1*, and *FAR2*, were correlated with SBP, DBP, and HTN. A total of 66 BP-correlated genes were identified from the analysis that did not adjust for cell types (Supplementary Dataset S2). Fifty-five genes (66%) identified in the adjusted analysis overlapped with those from the unadjusted analysis.

## Construction of coexpression networks and identification of BP-associated gene coexpression modules

To characterize BP-associated transcriptomic changes at a global level, we investigated coexpression patterns of multiple genes. We first constructed gene coexpression networks from gene expression data after adjusting for age, sex, BMI, cell types, technical covariates, and cohort to identify gene coexpression network modules (coEMs). Subsequently, we correlated the coEMs with BP values to capture the genomic coregulatory structure associated with BP variability. We identified 27 coEMs (Fig 2A; Supplementary Fig S2; the names of the coEMs are represented by different colors). Six coEMs (Turquoise, Blue, Red, Purple, Lightyellow, and Chocolate modules) were associated with either SBP or DBP at $P < 0.05$ and passed FDR $< 0.2$ (Fig 2B; Supplementary Fig S3). The Chocolate module passed FDR $< 0.05$. The Chocolate and Red modules were significantly enriched for genes in the BP signature gene set (*P*-values are 6.3e-12 and 0.03, respectively; Fig 2C).

We also constructed coexpression networks using the data that were unadjusted for cell types (Supplementary Fig S2). We identified 32 coEMs, of which the eigengenes of seven coEMs (Green, Greenyellow, Cyan, Magenta, Tan, Midnigtblue, and Lightgreen modules) showed significant correlation with SBP or DBP at $P < 0.05$. The Purple and Chocolate coEMs from the cell type-adjusted network were conserved with the Green and Lightgreen coEMs from the cell type-unadjusted results, respectively. The Turquoise coEM from the adjusted analysis split to three modules in the unadjusted analysis. Other coEMs were unique to the adjusted or unadjusted analysis (Supplementary Fig S4).

## Gene ontology (GO) enrichment analysis

To understand the biological pathways and functional categories of the BP transcriptomic changes, we conducted GO enrichment analysis of the BP signature gene set and the BP-correlated coEMs identified above. GO enrichment analysis for the BP signature gene set and coEMs from the cell type-adjusted analysis showed that the 83-gene BP signature gene set did not show any significantly enriched GO terms at Bonferroni-corrected $P < 0.05$. The top suggestive GO term enriched in the 83-gene signature set was apoptotic process ($P = 1.4$e-3). The Turquoise coEM is enriched with genes involved in chromatin modification (Bonferroni-corrected [BF] $P = 3.8$e-14), intracellular transport (BF $P = 1.1$e-11), and regulation of gene expression (BF $P = 5.9$e-11). The Purple coEM was enriched for hemostasis (BF $P = 5.0$e-4), platelet activation (BF $P = 1.8$e-3), and wound healing (BF $P = 7.5$e-3). The Chocolate coEM was enriched for immune cell-mediated cytotoxicity (BF $P = 2.6$e-8), cellular defense response (BF $P = 3.9$e-6), and inflammatory response (BF $P = 1.3$e-5) (Table 2). These results suggest that genes involved in multiple biological processes are tightly coregulated in relation to BP.

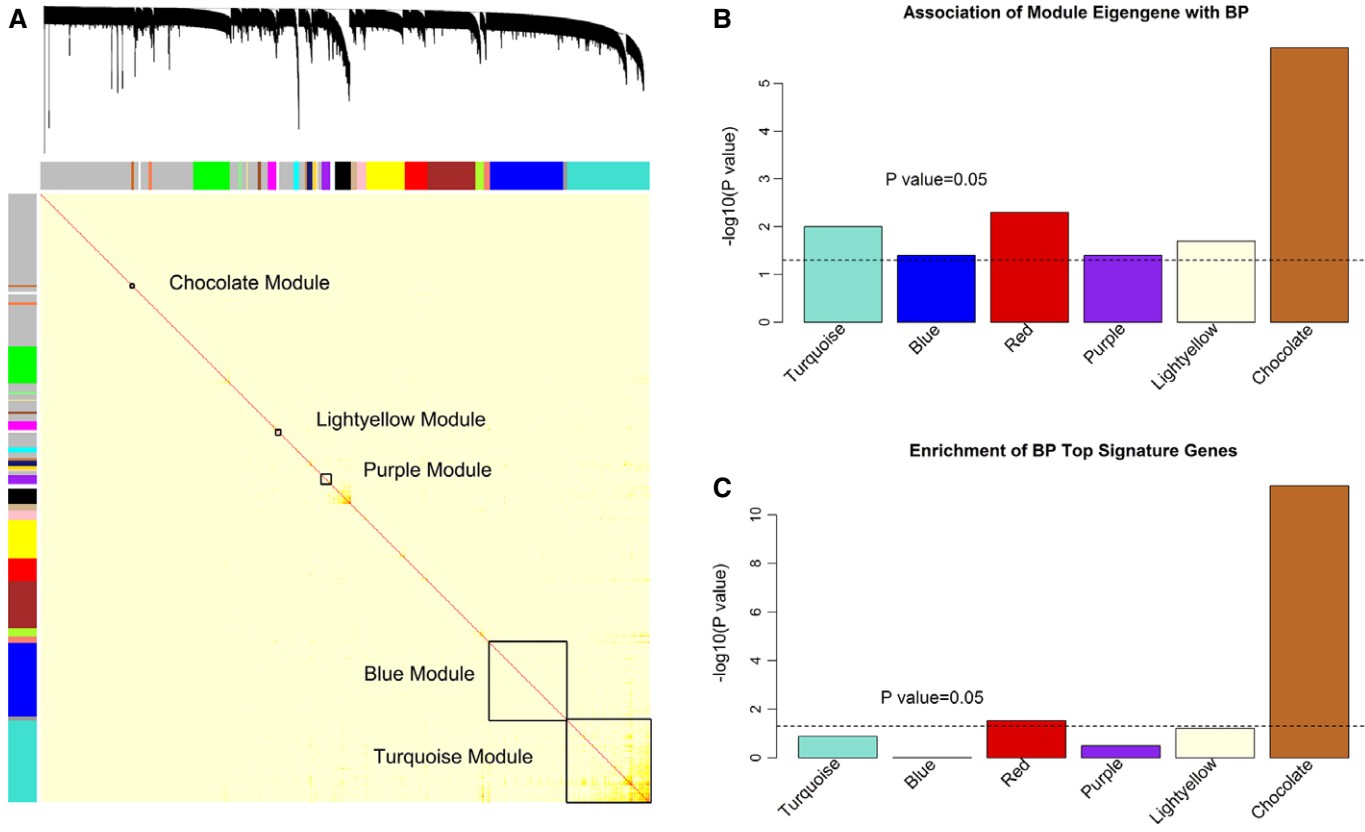

**Figure 2.  Identification of BP-associated coexpression network modules (coEMs).**

A   The topological overlap matrix (TOM) plot of coexpression network identified from gene expression profiling in 3,679 FHS participants. The six BP coEMs were highlighted.

B   The associations of eigengenes of each BP coEM with BP phenotypes (SBP or DBP). The *y*-axis is $-\log_{10}$-transformed *P*-value for the minimum association between the eigengenes of the module and BP traits (systolic or diastolic BP).

C   The enrichment of BP top signature genes in each BP coEM. The *y*-axis is $-\log_{10}$-transformed the enrichment *P*-value.

GO enrichment analysis for the coEMs from the cell type-unadjusted analysis showed significant enrichment of similar sets for biological processes or pathways as revealed by the coEMs in the adjusted analysis (Supplementary Table S2), suggesting that most of the BP coEMs and their represented biological process/pathways were relatively stable and were not affected by adjustment for cell counts.

**Inferring causal modules using SNP set enrichment analysis (SSEA)**

The BP signature genes and coEMs identified above could either play a causal role in regulating BP or be reactive to or independent of BP change. To differentiate these relationships, we used SSEA to evaluate whether the BP gene sets demonstrated enrichment for BP GWAS signals. For each BP-correlated gene set, we first retrieved blood eSNPs that showed association with the blood expression levels of genes in each BP-correlated gene set, and then extracted the BP association *P*-values of the eSNPs in the ICBP GWAS. Lastly, the overall distribution of the BP association *P*-values of the eSNPs representing each BP gene set was compared to the null distribution of all blood eSNPs using two statistical tests, the

Kolmogorov–Smirnov (KS) test and Fisher's exact test, to test whether the given BP gene set showed significant overall enrichment for eSNPs with stronger BP associations (see the Materials and Methods section). A BP gene set showing significance in SSEA is referred to as 'genetically inferred causal', because it is supported by orthogonal genetic evidence (i.e., association of its eSNPs with BP in GWAS) that is unlikely to be confounded by nongenetic factors. The same term also implies that further experimental validation is needed to establish causality with certainty.

Four coEMs from the cell count-adjusted network (Turquoise, Blue, Red, and Chocolate coEMs) were identified as genetically inferred causal gene sets at Bonferroni-corrected $P < 0.05$ by both KS and Fisher's exact tests (Table 3). The BP signature gene set, however, did not show enrichment for BP GWAS eSNPs. To explore which top genes contributed to the overall enrichment for BP-related genetic variants in the four genetically inferred causal coEMs, we retrieved the ICBP GWAS *P*-values for the blood eSNPs within these gene sets. We found that the blood eSNPs of 15 genes in the genetically inferred causal coEMs reached $P < 5e\text{-}8$ in the ICBP BP GWAS (Ehret *et al*, 2011) (Table 4). For example, rs3184504 (associated with BP in GWAS) is located in the third exon of *SH2B3* and is associated with the expression of four genes in the genetically inferred

**Table 2. Gene ontology enrichment analysis of the BP coexpression modules.**

| Gene set | Biological process terms | Gene count | Fold change | *P*-value | Bonferroni-corrected *P* |
|---|---|---|---|---|---|
| Turquoise | Chromatin modification | 89 | 2.5 | 4.6e-17 | 3.8e-14 |
| | Intracellular transport | 156 | 1.8 | 1.3e-14 | 1.1e-11 |
| | Regulation of gene expression | 382 | 1.4 | 7.1e-14 | 5.9e-11 |
| Purple | Hemostasis | 14 | 5.1 | 6.1e-7 | 5.0e-4 |
| | Platelet activation | 9 | 7.8 | 2.2e-6 | 1.8e-3 |
| | Wound healing | 14 | 4.1 | 9.1e-6 | 7.5e-3 |
| Chocolate | Immune cell-mediated cytotoxicity | 7 | 54.9 | 3.1e-11 | 2.6e-8 |
| | Cellular defense response | 10 | 12.4 | 4.7e-9 | 3.9e-6 |
| | Inflammatory response | 14 | 6.3 | 1.6e-8 | 1.3e-5 |

**Table 3. SNP set enrichment analysis of BP coexpression modules and BP signature gene set.**

| Module | SBP GWAS | | | | DBP GWAS | | | |
|---|---|---|---|---|---|---|---|---|
| | KS *P* | Permutation-based KS *P*[a] | Fisher *P* | Permutation-based Fisher *P*[a] | KS *P* | Permutation-based KS *P* | Fisher *P* | Permutation-based Fisher *P*[a] |
| BP signature | 0.98 | 0.96 | 1 | 1 | 0.20 | 0.23 | 1 | 1 |
| Turquoise | 2.8e-45 | < 0.001 | 7.8e-115 | < 0.001 | 1.8e-28 | < 0.001 | 3.0e-39 | < 0.001 |
| Blue | 1.4e-44 | < 0.001 | 7.0e-54 | < 0.001 | 1.3e-8 | < 0.001 | 3.4e-15 | < 0.001 |
| Red | 8.0e-5 | < 0.001 | 1.7e-17 | < 0.001 | 2.2e-15 | < 0.001 | 6.7e-19 | < 0.001 |
| Purple | 0.65 | 0.71 | 0.58 | 0.61 | 1 | 1 | 1 | 1 |
| Lightyellow | 1.6e-3 | 0.004 | 1 | 1 | 0.12 | 0.16 | 1 | 1 |
| Chocolate | 2.3e-14 | < 0.001 | 5.0e-5 | < 0.001 | 0.07 | 0.06 | 1 | 1 |

[a]Permutation-based *P* is empirically derived based on 1,000 permutations (see Materials and Methods). < 0.001 indicates none of the 1,000 random gene sets of matching size had *P*-values lower than the observed test *P*-values.

causal BP gene sets, including three genes (*SH2B3*, *ALDH2*, and *NAA25*) in *cis* and two genes (*IL8* and *TAGAP)* in *trans*. In addition, there are 34 additional genes whose *cis*- or *trans*-eSNPs (390 eSNPs in total) are associated with BP in the ICBP BP GWAS (Ehret *et al*, 2011) at *P* < 1e-5 (Supplementary Dataset S3). The SSEA results indicate that many genes in these genetically inferred causal BP gene sets carry a number of eSNPs (from whole blood profiling) with strong or moderate effects on BP regulation.

We also performed SSEA on the BP coEMs identified in the gene expression data without adjusting for cell types (Supplementary Table S3). The corresponding coEMs from the unadjusted analysis that significantly overlap with the Turquoise and Chocolate coEMs from the cell type-adjusted analysis remained significant in the SSEA analysis. Two other coEMs unique to the unadjusted analysis were found to be additional signals demonstrating enrichment for BP eSNPs (Supplementary Table S3).

**Identification of key drivers (KDs)**

Recent studies have shown that disease genes (or functionally correlated genes) are not distributed randomly in cellular or molecular interaction networks (Goh *et al*, 2007). Therefore, the graphic structure of the corresponding network models may help prioritize candidate genes for disease (Barabási *et al*, 2011; Huan *et al*, 2013; Zhang *et al*, 2013; Mäkinen *et al*, 2014). Based on this assumption,

we took advantage of pre-compiled blood Bayesian networks (BNs) (Emilsson *et al*, 2008) and protein–protein interaction (PPI) networks (Keshava Prasad *et al*, 2009) to identify key drivers (KDs) of the genetically inferred causal BP gene sets identified by SSEA. A KD is referred to as a local network hub whose neighbors in its local subnetwork show enrichment for BP genes in the genetically inferred causal gene sets. Due to their central location in the networks, KDs may have broad impact on multiple genes related to BP.

We projected the genes from each of the four genetically inferred causal BP coEMs constructed form cell count-adjusted results onto the blood BNs and the PPI network, and identified KDs using a KD analysis (Huan *et al*, 2013; Zhang *et al*, 2013), as detailed in the Materials and Methods section. Briefly, we took each gene in a given network as a candidate KD and tested whether the network neighborhood of the candidate KD was enriched for gene members of the genetically inferred causal BP gene set using Fisher's exact test. At Bonferroni-corrected *P* < 0.05, we identified 545 KDs from the HPRD PPI network (Keshava Prasad *et al*, 2009) and 131 from the blood BN (Supplementary Dataset S4). We further tested the reliability of these PPI KDs using an independent PPI database, BioGrid (Chatr-Aryamontri *et al*, 2013). We found that even though the direct overlap in PPIs between the two PPI databases was limited (15%), a considerably larger proportion of KDs replicated: 36% of KDs identified in HPRD replicated in BioGrid for the four genetically

**Table 4.** Genes in the genetically inferred causal BP gene sets whose blood eSNPs show significant association with BP in GWAS at $P < 5e-8$.

| SNP (Genomic location) | SNP Chr | ICBP GWAS SBP P-value | ICBP GWAS DBP P-value | cis or trans | Gene symbol | Gene chr | Gene set |
|---|---|---|---|---|---|---|---|
| rs3184504 (Coding, SH2B3)[a] | chr12 | 9.3e-10 | 2.3e-14 | cis | ALDH2 | chr12 | Turquoise |
| | | | | | SH2B3 | chr12 | Turquoise |
| | | | | | NAA25 | chr12 | Blue |
| | | | | trans[b] | IL8 | chr4 | Turquoise |
| | | | | | TAGAP | chr6 | Blue |
| rs3742004 (3UTR, FAM109A) | chr12 | 1.0e-6 | 2.2e-8 | cis | ATXN2 | chr12 | Turquoise |
| rs17367504 (Intron, MTHFR) | chr1 | 2.1e-10 | 1.3e-8 | cis | CLCN6 | chr1 | Turquoise |
| rs17249754 (Coding, ATP2B1) | chr12 | 9.7e-13 | 5.3e-9 | cis | GALNT4 | chr12 | Blue |
| rs198846 (3downstream, HIST1H1T) | chr6 | 2.2e-5 | 3.8e-8 | cis | HIST1H4B | chr6 | Turquoise |
| | | | | | BTN3A2 | chr6 | Turquoise |
| | | | | | HIST1H4C | chr6 | Turquoise |
| | | | | | HIST1H2BF | chr6 | Turquoise |
| | | | | | HIST1H4F | chr6 | Turquoise |
| | | | | | HIST1H3B | chr6 | Blue |
| rs17115100 (Intron, CYP17A1) | chr10 | 9.2e-10 | 1.4e-5 | cis | SFXN2 | chr10 | Blue |

[a]A proxy SNP rs653178 ($r^2 = 1$ with rs3184504) showing same cis- and trans-associations with genes listed for rs3184504. rs653178 is significantly associated with both SBP and DBP in ICBP GWAS, too (SBP $P = 9.3e-10$, and DBP $P = 1.6e-14$).
[b]The trans-associations between rs3184504 and those genes identified from Westra et al (2013).

inferred causal BP coEMs, and 50% of KDs in the Turquoise module replicated.

KDs were further ranked by leveraging their associations in BP GWAS, BP correlations, and their statistical significance in the KD analysis (see the Materials and Methods section). Table 5 lists the top 20 KDs, including seven genes having eSNPs associated with BP at $P < 1e-5$ in the ICBP BP GWAS (Ehret et al, 2011), 11 genes showing differential expression in relation to BP at transcriptome-wide significance, and two top KDs identified from both PPI and blood BNs.

**Inferring BP gene regulatory subnetworks driven by top key drivers**

We retrieved subnetworks derived from the top KDs in the blood BNs (Emilsson et al, 2008) and the HPRD PPI network (three examples are shown in Supplementary Fig S5). In Fig 3, we show subnetworks related to a top KD SH2B3 as a proof-of-concept. A missense SNP (rs3184504) located in an exon of SH2B3 (Fig 3A) was associated with BP and hypertension in GWAS (Ehret et al, 2011). rs3184504 has also been reported as a cis-eSNP for three genes and a trans-eSNP for sixteen genes in previous eSNP studies (Westra et al, 2013) (Joehanes et al, 2013a,b). Among all the candidate genes for BP identified in this study, we found 10 whose expression changes were associated with rs3184504 (two in cis and eight in trans), which constitute the 'SH2B3 genetic subnetwork' (Fig 3B). The 10 genes are as follows: SH2B3 (cis), a top BP KD; ALDH2 (cis), TAGAP (trans), and IL8 (trans), which are in the genetically inferred causal BP coEM (turquoise); ARHGEF40 (trans), TAGAP (trans), MYADM (trans), FOS (trans), PPP1R15A (trans), and S100A10 (trans), which are in the top BP signature gene set (green). The SH2B3-derived PPI subnetwork (Fig 3C) is enriched for genes involved in intracellular signaling cascade ($P = 1.2e-24$), T-cell activation ($P = 4.1e-12$), and T-cell differentiation ($P = 1.4e-9$). This SH2B3 subnetwork included STAT1, which is trans-associated with rs3184504, and two top BP signature genes KCNJ2 and PTPRO.

In order to systematically check whether the SH2B3-derived PPI subnetwork showed any enrichment for literature-based BP-related genes, we created a list of 657 BP-related genes by searching GeneRif (http://www.ncbi.nlm.nih.gov/gene/ about-generif; downloaded in Jan 2015) using the keywords 'hypertension' and 'blood pressure'. GeneRif includes literature descriptions of 14,069 unique human genes in total. We found that 41 of the 657 genes were present in the SH2B3-derived PPI subnetwork, which consisted of 362 genes in total, including PLCE1 (Ehret et al, 2011), BAT2 (Ehret et al, 2011), ADRB2 (Lou et al, 2010), RHOA (Connolly & Aaronson, 2011), and SOCS1 (Satou et al, 2012). Comparison of the two ratios 656/14,069 and 41/362 yielded $P = 5.5e-8$ (by the hypergeometric test) and 2.43-fold enrichment. This result indicates that the SH2B3-derived PPI subnetwork is enriched for known BP-related genes.

**Validation of the SH2B3 subnetworks in a $Sh2b3^{-/-}$ mouse model**

Although prior GWAS identified SH2B3 as a positional candidate gene, the role of SH2B3 in BP regulation remains unclear. As stated above, we identified SH2B3 as a putative KD for BP, and subnetworks based on SH2B3 revealed molecular interactions between this KD and many genes and multiple pathways related to BP regulation. In a related study (Saleh et al, 2015), we found that $Sh2b3^{-/-}$ mice had normal baseline BP but markedly elevated blood pressure in response to a low dose of angiotensin II (Ang II; 140 ng/kg/min) that did not affect BP in wild-type (WT) mice. This suggests a key role of Sh2b3 in BP regulation, and that loss or changes to this gene exacerbate response to hypertensive stimuli.

**Table 5. Top key drivers (KDs).**

| KD | Cellular network | | TWAS | GWAS | | |
|---|---|---|---|---|---|---|
| | KD *P*-value, corrected for subnetwork size | Tissue / network | *P*-value[a] for BP TWAS | eSNP ID | *P*-value[b] in ICBP GWAS | BP coEM |
| Top BP GWAS KDs | | | | | | |
| *SH2B3* | 4.4e-4 | HPRD | | rs653178 | 1.6e-14 | Turquoise |
| *ATXN2*[c] | 2.2e-5 | HPRD | | rs3742004 | 2.2e-8 | Turquoise |
| *NMT1* | 1.6e-5 | HPRD | | rs12946454 | 8.9e-8 | Turquoise |
| *NSF* | 5.0e-9 | HPRD | | rs17608766 | 7.3e-7 | Turquoise |
| *HSPA1B* | 2.1e-9 | HPRD | | rs805303 | 1.3e-6 | Blue |
| *BAT2* | 5.4e-7 | HPRD | | rs805303[d] | 1.3e-6 | Turquoise |
| *MAPKAPK5*[c] | 2.8e-8 | HPRD | | rs4767293 | 1.5e-6 | Turquoise |
| Top BP TWAS KDs | | | | | | |
| *GZMB*[c] | 2.0e-23 | Blood | 4.8e-22 | | | Chocolate |
| *PRF1* | 2.0e-26 | Blood | 2.5e-9 | | | Chocolate |
| *GPR56* | 1.2e-26 | Blood | 3.5e-9 | | | Chocolate |
| *RAB11FIP1* | 1.3e-3 | Blood | 4.0e-8 | | | Turquoise |
| *HIPK1*[c] | 3.1e-8 | HPRD | 9.1e-8 | | | Turquoise |
| *GZMH*[c] | 5.4e-24 | Blood | 3.3e-7 | | | Chocolate |
| *VIM* | 6.0e-3 | HPRD | 4.2e-7 | | | Turquoise |
| *BCL2L11*[c] | 1.8e-10 | HPRD | 1.7e-6 | | | Blue |
| *BHLHE40*[c] | 3.8e-7 | HPRD | 2.2e-6 | | | Turquoise |
| *KLRD1* | 1.2e-30 | Blood | 2.5e-6 | | | Chocolate |
| *TGFBR3* | 1.1e-26 | Blood | 2.5e-6 | | | Chocolate |
| Top multi-tissue/ network KDs | | | | | | |
| *DCLRE1C*[c] | 1.1e-16 | HPRD, Blood | | | | Blue |
| *ERCC6* | 8.8e-14 | HPRD, Blood | | | | Turquoise |

[a]*P*-values passing transcriptome-wide significance at Bonferroni-corrected *P* < 0.05 (corrected for 17,318 measured genes).
[b]Minimum *P*-values for SBP, DBP, and HTN associations.
[c]Indicating the KD could be replicated in BioGrid PPI database (Chatr-Aryamontri *et al*, 2013).
[d]*trans*-eSNP; other eSNPs are *cis*-eSNPs.
PPI, protein–protein interaction; HPRD, Human Protein Reference Database (Keshava Prasad *et al*, 2009).

To determine the accuracy of the predicted network structure, we performed RNA sequencing of the entire transcriptome in whole blood from WT and $Sh2b3^{-/-}$ mice. Whole blood was chosen to complement our network finding in whole blood from humans. As shown in Supplementary Fig S6, the RNA reads of exons 3–8 of $Sh2b3$ in the $Sh2b3^{-/-}$ mice were absent, in complete agreement with the design of the targeting construct used to produce these mice (Takaki *et al*, 2000). At a false discovery rate (FDR was estimated using the *q* value approach) < 0.05, we found 2,240 differentially expressed genes between WT and $Sh2b3^{-/-}$ mice (Supplementary Dataset S5), supporting a large-scale perturbation of the transcriptome in the $Sh2b3^{-/-}$ mice. The gene signatures showed significant enrichment for genes involved in immune response (*P* = 4.0e-22), inflammatory response (*P* = 3.5e-20), and T-cell activation (*P* = 4.1e-6), consistent with the pathways in the predicted SH2B3 subnetworks. More importantly, as shown in Table 6, these signature genes significantly overlapped with those in the SH2B3 genetic subnetwork (Fig 3B) at *P* = 1.2e-5 and the SH2B3 PPI subnetwork (Fig 3C) at *P* = 2.2e-14. The overlapping

genes between the SH2B3 subnetworks and the signatures observed in $Sh2b3^{-/-}$ mice (Supplementary Table S4) again showed significant enrichment for intracellular signaling cascade (*P* = 9.4e-16) and T-cell activation (*P* = 5.0e-6). These results strongly support our predicted SH2B3 subnetworks. Consistent with our prediction, Saleh *et al* (2015) also confirmed the exacerbation of inflammation and T-cell activation in $Sh2b3^{-/-}$ mice.

## Discussion

This study represents a large, single-site transcriptome-wide analysis of BP in 3,679 individuals who were not receiving antihypertensive drug treatment. Extending traditional transcriptome-wide analysis that targets differentially expressed genes at the individual-gene level, we also conducted higher level coexpression network analysis to identify multiple genes demonstrating coregulatory network structure in the form of coEMs associated with BP. To differentiate causal from reactive roles of the BP-related genes/gene

**Table 6.   Summary of the overlap between gene signatures of *Sh2b3*<sup>−/−</sup> mice and the predicted SH2B3 subnetworks.**

| SH2B3 subnetwork | Number of genes in the subnetwork | Number of overlapping genes | Fold enrichment | *P*-value |
|---|---|---|---|---|
| Genetic subnetwork | 19 | 8 | 2.5 | 1.2e-5 |
| PPI subnetwork | 362 | 78 | 1.3 | 2.2e-14 |

sets from transcriptome-wide analysis, we integrated the differentially expressed genes and the BP coEMs with SNP association results from BP GWAS. To further pinpoint key BP genes and dissect key regulatory mechanisms among the genetically inferred causal BP gene sets, we projected genes within these gene sets onto gene/protein networks and identified key drivers (KDs). These KDs appear to regulate a large number of interacting genes in gene subnetworks and orchestrate multiple biological processes and pathways underlying BP regulation.

By first applying a traditional approach that focused on differentially expressed individual genes, we identified a gene signature set comprised of 83 genes whose expression levels were correlated with BP traits. The 83-gene BP signature gene set did not show significant enrichment for biological processes or pathways suggesting that the traditional single-gene approach lacks power to capture high-order organization of genes underlying BP regulation. Subsequent SNP set enrichment analysis (SSEA), which incorporates genetic signals, did not support an overall causal role of the top BP signature genes. Although this lack of overall significance in SSEA does not exclude a small subset of genes being causal, for example, a top signature gene *ATP2B1* has been previously detected as a GWAS signal in ICBP GWAS at $P < 5 \times 10^{-8}$. *ATP2B1* (ATPase, Ca$^{2+}$ transporting,

plasma membrane 1) is known to be responsible for Ca$^{2+}$ transportation in plasma membrane, and a BP-associated ATP2B1 SNP has been linked to *ATP2B1* expression in umbilical artery smooth muscle cells (Tabara *et al*, 2010). Ca$^{2+}$ is critical for muscle contraction (Marks, 2003), and defects or altered expression of *ATP2B1* will likely induce changes in artery smooth muscle contraction which may in turn affect blood pressure variability. Another top signature gene, *FOS* (known as *c-fos*), has been found to be associated with hypertension (Minson *et al*, 1996; Cunningham *et al*, 2006). The *c-fos* gene is considered to be a useful marker of neuronal activity in different sites, including those important in BP control. In the rat, *c-fos* expression in the brain is likely to be important for BP control, and the blockade of *c-fos* expression in this region attenuates resting and stimulated BP levels. Inhibition of local neuronal activity acutely increased both BP and immunoreactivity to Fos, the protein product of the *c-fos* gene. Intravenous infusion of sodium nitroprusside induced hypotension, and the number of Fos-positive spinal sympathetic neurons increased (Minson *et al*, 1996). Several additional BP signature genes have been reported to be involved in BP-related diseases or processes such as cardiovascular disease (e.g., *ABCA1* (Tang & Oram, 2009), *AHR* (Zhang, 2011), and *GZMB* (Joehanes *et al*, 2013a,b)), type II diabetes (e.g., *ABCA1* (Tang &

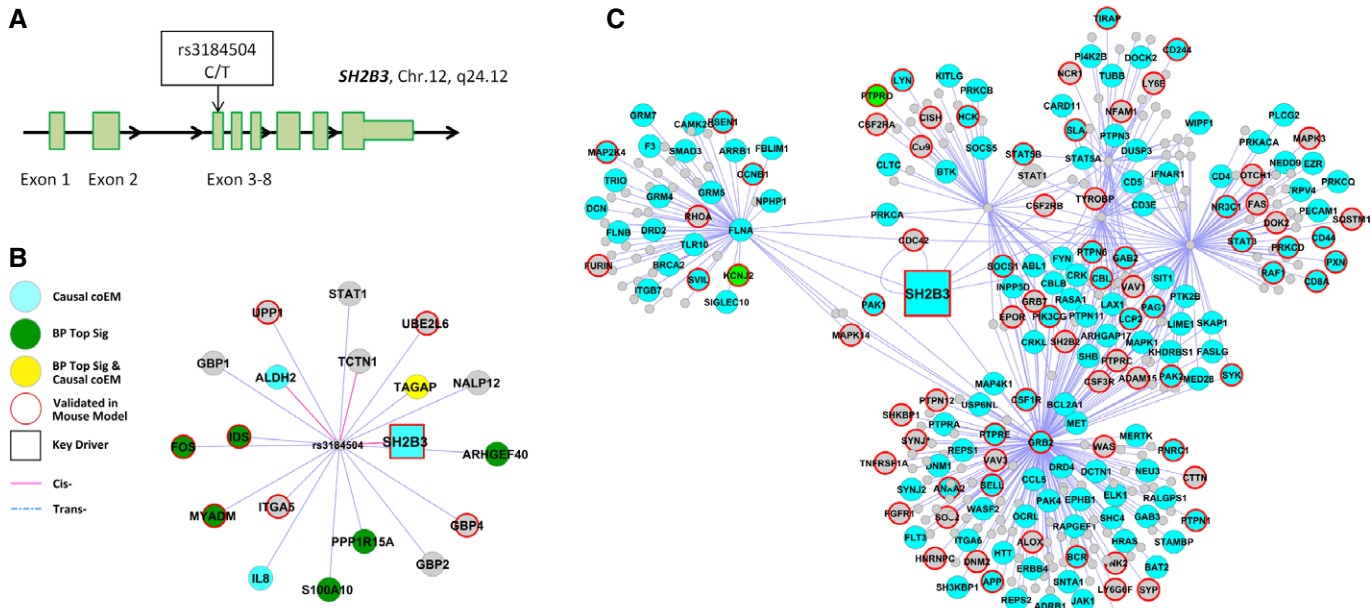

**Figure 3.   *SH2B3*-related genetic and protein–protein interaction subnetworks.**

A   rs3184504, a missense SNP, is located in the third exon of *SH2B3*.

B   *SH2B3* genetic subnetwork. rs3184504 is associated with 19 genes in a *cis* or *trans* manner based on analysis of eQTLs.

C   *SH2B3* protein–protein interaction (PPI) subnetwork. *SH2B3* is depicted as a rectangular node. Green nodes indicate differentially expressed BP genes at Bonferroni-corrected *P* < 0.05 in the Framingham Heart Study (FHS) data (BP Top Sig); turquoise nodes indicate genes in the BP causal coEMs; yellow nodes indicate genes that are in both the BP Top Sig set and the BP causal coEMs. The nodes marked with a red border indicate differentially expressed genes between wild-type (WT) and *Sh2b3*<sup>−/−</sup> mice.

Oram, 2009), *ANXA1* (Lindgren *et al*, 2001), and *PTGS2* (Shanmugam *et al*, 2006)), and inflammation (e.g., *GZMB* (Hiebert & Granville, 2012) and *KLRD1* (Choi *et al*, 2012)). We speculate that these genes may play important roles in BP regulation, but further mechanistic studies are necessary.

To overcome the inherent limitations of single-gene-based analysis, we used a coexpression network approach to capture high-order gene–gene interrelation in association with BP that cannot be revealed by single-gene-based analysis. We chose to use WGCNA (Zhang & Horvath, 2005; Langfelder & Horvath, 2008) to construct a gene coexpression network; this approach has revealed patterns associated with other diseases (Chen *et al*, 2008; Emilsson *et al*, 2008; Huan *et al*, 2013; Zhang *et al*, 2013). The superior performance of WGCNA in detecting biologically coherent gene coregulation structures is built upon its underlying algorithm that takes into consideration not only pairwise gene–gene correlations but also the similarity between a given pair of genes in their correlation structure with the rest of the genome. Incorporation of the higher order relationships in WGCNA yields more coherent coexpression modules compared to classic clustering approaches. Indeed, a majority of the WGCNA modules identified in our study contain functionally related genes. For instance, the Turquoise module is highly enriched for genes involved in transcriptional regulation and chromatin modification; the Chocolate module mainly contains genes important for inflammation and immune response.

Of note, the known functional information was not inputted into network construction; instead, the network modules were purely defined by our blood transcriptome data. The coherence between the coexpression patterns and functional relatedness of the module genes supports the power and accuracy of WGCNA. Further, four of the six modules correlated with BP tested as being causal by genetic inference from SSEA. This is in contrast to the lack of SSEA significance for the BP signature gene set and highlights the capacity of WGCNA to unravel genes and processes that are more likely upstream of BP variation. Moreover, there exists literature support for the biological plausibility of the genetically inferred causal modules detected. For instance, both the Chocolate and Turquoise modules were found to be causally linked to BP regulation. The causal links between immune/inflammatory response represented by the Chocolate module and hypertension have been shown in numerous studies (Harrison *et al*, 2010, 2011, 2012; Barhoumi *et al*, 2011). The epigenetic and transcriptional regulations, represented by the Turquoise module, are key processes implicated for many diseases (Maunakea *et al*, 2010). Recent literature supports the notion that key hypertension genes and pathways, such as the $Na^+$-$K^+$-$2Cl^-$-cotransporter 1 (*NKCC1*) (Cho *et al*, 2012) and angiotensin-converting enzyme 1 (*ACE1*) (Lee *et al*, 2012), are epigenetically regulated in relation to hypertension. These lines of evidence support the successful application of coexpression network approaches in this study.

The genetically inferred causal coEMs identified for BP through our network analysis contain a large number of genes that are tightly coregulated, which creates a challenge in finding the 'key' regulatory genes and dissecting gene–gene interactions that might provide insight into clinical treatment of hypertension. To address this limitation, we employed graphical molecular networks including Bayesian networks constructed from blood-derived transcriptomic data (Emilsson *et al*, 2008) and PPI networks (Keshava Prasad *et al*, 2009; Chatr-Aryamontri *et al*, 2013) that provide detailed topological information regarding gene–gene and protein–protein relations to tease out regulatory cascades and prioritize key regulatory genes for BP. We used a network-based gene ranking method, key driver (KD) analysis approach, to identify key drivers in each genetically inferred causal coEM. Previous studies have shown that KD analysis has the capacity to uncover novel genes that play important roles in disease development but are missed by traditional approaches (Wang *et al*, 2012; Huan *et al*, 2013; Zhang *et al*, 2013; Mäkinen *et al*, 2014). Many of the KDs that we identified have previously been reported to be involved in BP regulation, including *WNK1* (Choate *et al*, 2003), *BMPR2* (Atkinson *et al*, 2002; Hamid *et al*, 2009), *GPX1* (Ardanaz *et al*, 2010), *TAF1* (Koschinsky *et al*, 2001), *GYS1* (Groop *et al*, 1993; Orho-Melander *et al*, 1999), *CAST* (Kokubo *et al*, 2006), *IKBKAP* (Kokubo *et al*, 2006), *MEF2A* (Oishi *et al*, 2010), and *PPARA* (Bernal-Mizrachi *et al*, 2003, 2007), supporting the validity of our methods. However, as KDA relies on precomputed gene–gene interaction networks and topology-based gene ranking, its performance is highly dependent on the quality of the molecular networks used. In particular, the currently available human PPI networks are prone to high false positives (Cusick *et al*, 2005). Therefore, additional prioritization based on orthogonal evidence and explicit experimental validation is warranted. By implementing multiple criteria to rank KDs, we were able to focus on a subset of KDs that are more likely to be true signals, one of which is *SH2B3*.

*SH2B3* was identified as a top KD ($P = 4.4e-4$) in the PPI network (Keshava Prasad *et al*, 2009). A missense SNP (rs3184504) located in the third exon of *SH2B3* was reported to be associated with BP and hypertension in prior GWAS (Ehret *et al*, 2011). However, the molecular mechanisms relating *SH2B3* to BP regulation have not previously been reported. Our network analysis predicted that the SH2B3-related PPI subnetwork is highly enriched for genes involved in intracellular signaling and T-cell activation and differentiation. Through transcriptome-wide sequencing of whole-blood-derived RNA from WT and $Sh2b3^{-/-}$ mice, we found that our predicted SH2B3-derived genetic and PPI subnetworks overlapped greatly with the differentially expressed genes in $Sh2b3^{-/-}$ versus WT mice, thus validating our predicted networks derived from humans. In our related paper (Saleh *et al*, 2015), we confirmed that deletion of *Sh2b3* exacerbates Ang II-induced hypertension via mechanisms involving inflammation and T-cell activation. These results are consistent with previous evidence linking *SH2B3* to a range of signaling cascade activities such as cytokine signaling (Takizawa *et al*, 2008) as well as recent findings on the role of T lymphocytes in hypertension through a feed-forward mechanism: modest degrees of blood pressure elevation lead to T-cell activation, which in turn promotes inflammation and further blood pressure elevation (Marvar *et al*, 2010; Kirabo *et al*, 2014). Mice lacking T lymphocytes are resistant to the development of both Ang II- and DOCA-salt-induced hypertension (Guzik *et al*, 2007). Our findings therefore demonstrate the utility of our systems biology approach to identify not only candidate genes such as *SH2B3* but also gene network-level mechanism through which the adaptor protein SH2B3 contributes to hypertension through perturbation of inflammatory and T-cell functions.

In conclusion, our integrative and systems biology analysis, which leveraged transcriptional profiling, GWAS, and network modeling, revealed multiple biological processes that contribute to BP regulation. This approach highlighted putative regulatory roles of key driver genes, most notably *SH2B3*. The key drivers and the

biological processes they regulate form coherent gene–gene regulatory networks. Future follow-up studies focusing on the novel KDs and their network structures are warranted to provide insights into the complex mechanisms underlying BP pathophysiology.

## Materials and Methods

### Study participants

Biosamples were collected, and BP measurements were obtained from all available Framingham Heart Study (FHS) offspring cohort participants who attended their eighth clinic examination (2005–2008; $n = 2,446$) (Feinleib *et al*, 1975) and third-generation cohort participants who attended their second examination (2008–2011; $n = 31,80$) (Splansky *et al*, 2007). Data from 3,679 participants (offspring cohort, $n = 1,102$; third-generation cohort, $n = 2,577$) who were not receiving antihypertensive drugs were used for this project. Hypertension (HTN) was defined as SBP ≥ 140 mm Hg or DBP ≥ 90 mm Hg. This study followed the recommendations of the Declaration of Helsinki and the Department of Health and Human Services Belmont Report, and is approved under the Boston University Medical Center's protocol H-27984. Informed consent was obtained from each participant.

### Gene expression profiling measurement

Whole blood samples (2.5 ml) were collected in PAXgene™ tubes (PreAnalytiX, Hombrechtikon, Switzerland) during Framingham offspring cohort examination 8 (2005–2008) and during Framingham third-generation cohort examination 2 (2008–2011). Total RNA was isolated from frozen PAXgene blood tubes by Asuragen, Inc., according to the company's standard operating procedures. The purity and quantity of total RNA samples were determined by absorbance readings at 260 and 280 nm using a NanoDrop ND-1000 UV spectrophotometer. Expression profiling was carried out on samples that passed RNA quality control. RNA expression was conducted using the Affymetrix Human Exon Array ST 1.0 (Affymetrix, Inc., Santa Clara, CA). All data used herein are available online in dbGaP (http://www.ncbi.nlm.nih.gov/gap; accession number phs000007).

### Gene expression data normalization

The raw gene expression data were at first preprocessed by quartile normalization. Then, the RMA (robust multi-array average) values of every gene (17,318 measured genes) were adjusted for a set of technical covariates (e.g., batch, details in Supplementary Table S5) by fitting linear mixed regression (LME) models. Imputed blood cell counts (i.e., white blood cell [WBC], red blood cell [RBC], platelet, lymphocyte, monocyte, eosinophil, and basophil) (Joehanes R, in preparation) were also evaluated as covariates and adjusted if deemed significant, as detailed below. The residuals were retained for further analysis.

### Influence of blood cell types on mRNA expression

Gene expression levels were measured from whole blood, which contains multiple cell types. We used LME models to test the associations between cell types and transcript levels. A large proportion of transcripts were found to be associated with cell-type proportions. The top three cell types are white blood cells, neutrophils, and lymphocytes. To further evaluate how differences in cell-type proportions affect the BP-associated genes at the level of single genes and the coexpression modules identified in this study, we conducted our overall analysis both with and without accounting for cell-type effects to capture both cell type-dependent and cell type-independent BP-associated genes and processes. We report both results but focus our discussions on those from the adjusted analyses.

### Identification of differentially expressed genes for BP traits

The association between gene expression residuals (see in 'Gene expression data normalization') and BP traits (SBP, DBP, and HTN) was tested by a linear mixed model adjusting for age, sex, MI, cell types, and familial relatedness in the FHS families. As a secondary analysis, the association between gene expression residuals and BP traits was tested by a linear mixed model adjusting for age, sex, BMI, and familial relatedness, but not adjusting for cell types. Association analysis was conducted using the Kinship package in R (http://cran.r-project.org/web/packages/kinship/) (Abecasis *et al*, 2001). The BP signature genes were chosen at Bonferroni-corrected $P < 0.05$ (corrected for 17,318 measured genes).

### Constructing gene coexpression networks and identifying BP-associated coexpression network modules

Gene coexpression networks were constructed using weighted gene coexpression network analysis (WGCNA) (Zhang & Horvath, 2005; Langfelder & Horvath, 2008). The WGCNA R package uses a fitting index to evaluate a scale-free network structure built upon Pearson's gene–gene correlations from gene expression variances among individuals (Zhang & Horvath, 2005). Genes were grouped based on the topological overlap of their connectivity using average linkage hierarchical clustering (Zhang & Horvath, 2005), followed by a dynamic cut-tree algorithm to dynamically cut the clustering dendrogram branches into gene coexpression network modules (coEMs) (Langfelder *et al*, 2008).

The coexpression network was constructed using the gene expression data from all 3,679 individuals who were not on antihypertensive treatment, rather than on normotensive and hypertensive individuals separately. Inclusion of all individuals across the full spectrum of BP variability increases our power to capture coregulated genes associated with BP variability. In order to minimize the effects from other covariates of BP that may affect the network structure, the gene expression data were pre-adjusted for BP covariates including age, sex, BMI, cell types, and cohort (offspring vs. third-generation). The residuals were kept for the coexpression network construction (see 'Gene expression data normalization'). First, we built weighted gene coexpression networks and identified coEMs that fit a scale-free topological structure by fitting the index $R^2 > 0.8$ of the linear model that regressed $log(p(k))$ on $log(k)$, where $k$ is the connectivity of every node (gene) in the network and $p(k)$ is the frequency distribution of connectivity. The fitting index of a perfect scale-free network is 1. The relations of coEMs to BP phenotypes were evaluated by correlating the eigengene (first principle

component) of each coEM with SBP and DBP across all 3,679 participants via Pearson's correlation testing; $P < 0.05$ was considered significant.

## Identification and collection of whole blood eSNPs

The eSNPs of whole blood used in this study were combined from: (1) eSNPs identified from FHS whole blood gene expression (~18,000 genes) and genotype data (~8 million SNPs after imputation to the 1,000 Genomes reference panel) (Joehanes R, in preparation), and (2) eSNPs identified from other published resources (Emilsson *et al*, 2008; Fehrmann *et al*, 2011; Lappalainen *et al*, 2013; Westra *et al*, 2013; Battle *et al*, 2014; Wright *et al*, 2014).

The FHS blood eQTLs were generated using data from 5,257 FHS participants with genome-wide genotype data and gene expression profiling. DNA isolation, and genotyping with the Affymetrix 500K mapping array and the Affymetrix 50 K gene-focused MIP array have been described previously (Levy *et al*, 2009). Imputation of ~36.3 million SNPs in 1,000 Genomes Phase 1 SNP data was conducted using MACH (Li *et al*, 2010). For the eQTL identification, we used the 1,000 Genome resource-imputed SNPs with minor allele frequency (MAF) > 0.01 and imputation ratio > 0.3, yielding approximately 8 million SNPs for eQTL analysis. A pedigree-based linear mixed model was used to determine the association between each gene expression value and the imputed SNP genotypes by adjusting for age, sex, technical covariates, cell types, and familial relatedness. The *cis*-eSNPs (or eQTLs) were constrained by a 1-megabyte (Mb) window on either side of the transcription start site (TSS). The remaining eSNPs were defined as *trans*-eSNPs. Genomic coordinates were based on NCBI human reference genome build 37/hg19. The Benjamini–Hochberg method (BH) (Benjamini & Hochberg, 1995) was used to calculate false discovery rates (FDR) of *cis*- and *trans*-eQTLs separately. We only considered eQTLs (both *cis* and *trans*) at FDR < 0.1 in each of the blood eQTL studies.

## SNP set enrichment analysis (SSEA)

SSEA was used to determine whether a group of eSNPs corresponding to a gene set was enriched for SNPs with low BP association *P*-values in GWAS (Zhong *et al*, 2010; Mäkinen *et al*, 2014). A gene set that passed SSEA testing was considered as a potential causal gene set. The GWAS used in the current investigation was from the International Consortium of Blood Pressure (ICBP) GWAS for SBP and DBP (Ehret *et al*, 2011). We first mapped genes within the BP signature gene set or BP coEMs to eSNPs and retrieved the BP association *P*-value for every eSNP. Then, we refined the eSNPs of the gene set by only keeping one eSNP for eSNP pairs in linkage disequilibrium (LD, defined as pairwise $r^2 \geq 0.8$). The GWAS association *P*-values of this group of refined eSNPs were denoted as $P_{geneset}$. The deviation of $P_{geneset}$ from the null distribution of all eSNPs (limited to eSNPs with LD $r^2 < 0.8$) toward a lower *P*-value was evaluated by both the Kolmogorov–Smirnov (KS) test to derive $P_{KS}$ and Fisher's exact test to obtain $P_{Fisher}$. In order to perform Fisher's exact test, we categorized all eSNPs into significant and nonsignificant categories based on their association with BP using a BP GWAS *P* value threshold of $P < 0.05$.

For each gene set and each statistical test (KS or Fisher's exact test), we computed empirically derived enrichment *P*-values based

on 1,000 permutations. Each permutation involved random sampling of equal number of genes matching the gene set being tested. Each permutation gene set was subject to the same KS or Fisher's exact test as was used for the testing gene set. The empirically derived *P*-value was estimated as the number of permutation gene sets with *P*-values less than the observed *P*-value of a given gene set/1,000. The BP gene set or a coexpression module was considered to be genetically inferred causal for BP if it demonstrated overall enrichment with GWAS eSNPs showing low *P*-value associations with SBP or DBP at Bonferroni-corrected $P_{KS}$ and $P_{Fisher} < 0.05$ (corrected for seven gene sets including one BP signature gene set and six BP coEMs).

## Molecular network models from orthogonal studies

We compiled both literature-based and data-driven molecular networks including gene regulatory networks and protein–protein interaction (PPI) networks. The data-driven networks were Bayesian networks (BNs) constructed from blood tissue from a human dataset (Emilsson *et al*, 2008) using a BN modeling method (Zhu *et al*, 2004, 2007). The BN modeling methods deduce gene regulatory network structure in the transcriptome-wide gene expression data using a Markov chain Monte Carlo (MCMC) approach and use genetic data as prior information to infer directionality between genes (Zhu *et al*, 2004, 2007).

The literature-driven network used in the current study is the PPI network downloaded from the Human Protein Reference Database (HPRD) (Keshava Prasad *et al*, 2009). PPIs in HPRD were manually curated from literature by biologists. We also downloaded PPI network from BioGrid (Chatr-Aryamontri *et al*, 2013), also based on PPIs curated from literature as a comparison with HPRD.

## Key driver (KD) analysis

For the genetically inferred causal BP gene sets identified by SSEA, we integrated the genes with molecular networks (BNs and PPI networks as described above) to identify key regulators of every BP gene set using KD analysis. The objective of KD analysis was to identify the important genes for a gene set with respect to a given network structure. A KD of a BP causal gene set is defined as a gene whose neighbor genes in the network are significantly enriched for genes in the BP gene set. As illustrated in Supplementary Fig S7, in order to test whether gene *G* in a network (a BN or PPI network) is a KD or not, first, we identified the subnetwork of *G* by retrieving its directly connected genes (1st-layer neighbor genes), the genes connected by its 1st-layer neighbor genes (2nd-layer neighbor genes), and the genes connected by its 2nd-layer neighbor genes (3rd-layer neighbor genes). Next, we used Fisher's exact test to evaluate whether the genes in the subnetwork of *G* (1st- to 3rd-layer neighbor genes of *G*) show enrichment for genes in the BP causal gene set to derive a KD-enrichment *P*-value. A *G* that reached a Bonferroni-corrected KD-enrichment $P < 0.05$ was reported as a KD (after correction for the number of genes in the 3rd-layer expanding network of the tested BP causal gene set).

After the identification of KDs for each BP causal coEMs in each network (BNs and PPI network), KDs were further ranked by leveraging: (1) the BP association *P*-values of the eSNPs in the KD based on results from the ICBP BP GWAS (Ehret *et al*, 2011); (2) the

differential expression association *P*-value for BP from transcriptome-wide differential expression analysis in the single-gene level; and (3) the KD-enrichment *P*-values.

### Gene ontology analysis

Each BP gene set identified in this study was classified using Gene Ontology (GO)—biology process categories to define biological process enrichment (Ashburner *et al*, 2000). One-sided Fisher's exact test was performed to calculate enrichment *P*-values. The *P*-value was further corrected by the number of unique GO biological process terms ($N = 825$). A threshold of $P < 6e{-}5$ was considered significant.

### Mouse models for validation of *Sh2b3*

All animal procedures were approved by Vanderbilt University's Institutional Animal Care and Use Committee, and mice were housed and cared for in accordance with the Guide for the Care and Use of Laboratory Animals. Wild-type (WT) C57Bl/6J mice were purchased from Jackson Laboratories. *Sh2b3*-deficient mice ($Sh2b3^{-/-}$) were provided by Dr. Satoshi Takaki (International Medical Center of Japan, Tokyo, Japan) and generated as previously described (Takaki *et al*, 2000). Exons 3–8 of the gene were knocked out by a genetic recombination technology. These mice were back-crossed with C57Bl/6J for > 10 generations.

### RNA sequencing of whole blood samples from mouse models

RNA samples from whole blood were extracted from 4 WT and 4 $Sh2b3^{-/-}$ mice. Mouse blood total RNA was isolated using a RiboPure™ RNA Purification Kit (Cat# AM1928, Life Technologies, Carlsbad, CA) with an RNase-free DNase treatment per the manufacturer's instructions. cDNA library construction and RNA sequencing were performed by VANTAGE (Vanderbilt Technologies for Advanced Genomics, Vanderbilt University Medical Center, Nashville, TN). Library preparation was performed using the Illumina TruSeq Stranded mRNA Sample Preparation Kit, and Rev. D of the protocol. Samples were sequenced on the Illumina HiSeq 2500 using v3 SBS chemistry. Libraries were sequenced on a Single Read 50 bp run at 30 million passing filter reads/sample. Details of RNA sequencing protocols were available on: http://vantage.vanderbilt.edu/resources/grant-text/.

### RNA sequencing data analysis

Quality control (QC) of the RNA-Seq reads from FASTQ files was performed by FASTX-Toolkit package (http://hannonlab.cshl.edu/fastx_toolkit/), and 99.8% reads passed QC by filtering out reads of quality score < 28 and length < 30 bp. The RNA-Seq reads passing QC were mapped to the mouse reference genome (UCSC mm10) using Tophat v2.0 (Trapnell *et al*, 2009). Duplicated reads with identical external coordinates were further removed from the output binary sequence alignment (BAM) format files using SAMtools (samtools rmdup command) (Li *et al*, 2009). Cufflinks v2.2 was used to estimate and normalize mRNA abundances in the form of FPRM (expected number of fragments per kilobase of transcript sequence per millions base pairs sequenced) values (Trapnell *et al*, 2010). Cuffdiff (Trapnell *et al*, 2010) was used to identify

differentially expressed genes between WT and $Sh2b3^{-/-}$ groups. Multiple testing was corrected using the *q* value approach, and the significant threshold of differentially expressed genes was set to be *q* value < 0.05 (Storey & Tibshirani, 2003).

### Data availability

The gene expression data and phenotypes of Framingham Heart Study participants are available online in dbGaP (http://www.ncbi.nlm.nih.gov/gap; accession number phs000007). The RNA sequencing data of wild-type and $Sh2b3^{-/-}$ mice are accessible through the GEO accession number GSE65348.

**Supplementary information** for this article is available online: http://msb.embopress.org

### Acknowledgements

The Framingham Heart Study is funded by National Institutes of Health contract N01-HC-25195. The laboratory work for this investigation was funded by the Division of Intramural Research, National Heart, Lung, and Blood Institute, National Institutes of Health (Dr. Daniel Levy), and NIH grant K08HL121671 (Dr. Meena Madhur). The analytical component of this project was funded by the Division of Intramural Research, National Heart, Lung, and Blood Institute, the Center for Information Technology, National Institutes of Health, Bethesda, MD (Dr. Daniel Levy), and the American Heart Association Scientist Development Grant 13SDG17290032 (Dr. Xia Yang).

### Author contributions

DL and XY designed, directed, and supervised the project. TH, XY, and DL drafted the manuscript. MM, DH, MS, and NE designed and conducted knockout mouse experiments. NR, RW, and PL conducted the gene expression microarray experiments. TH, QM, DJ, BC, and SY conducted the analyses. DL, XY, JZ, BZ, AJ, CO, RV, and PM directed the design of analysis approaches. All authors participated in revising and editing the manuscript. All authors have read and approved the final version of the manuscript.

### Conflict of interest

The authors declare that they have no conflict of interest.

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
