## [Review Process File · Molecular Systems Biology]

Integration of Transcriptomic Profiling, Genome-wide Association, and Network Biology Reveals Molecular Mechanisms Underlying Blood Pressure Regulation

Tianxiao Huan, Qingying Meng, Mohamed A. Saleh, Allison E. Norlander, Roby Joehanes, Jun Zhu, Brian H. Chen, Bin Zhang, Andrew D. Johnson, Saixia Ying, Paul Courchesne, Nalini Raghavachari, Richard Wang, Poching Liu, The International Consortium for Blood Pressure GWAS (ICBP), Christopher J. O'Donnell, Ramachandran Vasani, Peter J. Munson, Meena S. Madhur, David G. Harrison, Xia Yang, and Daniel Levy

Corresponding author: Daniel Levy, FHeart Study

Review timeline:

Submission date:	28 April 2014
Editorial Decision:	11 June 2014
Revision received:	02 December 2014
Editorial Decision:	13 January 2015
Revision received:	10 February 2015
Editorial Decision:	02 March 2015
Revision received:	08 March 2015
Accepted:	10 March 2015

Editor: Maria Polychronidou

Transaction Report:

1st Editorial Decision

11 June 2014

Thank you again for submitting your work to Molecular Systems Biology. We have now heard back from the three referees who agreed to evaluate your manuscript. As you will see from the reports below, the referees acknowledge that the presented analysis is potentially interesting and that the gene expression data could serve as a useful resource. However, they raise a series of concerns, which should be carefully addressed in a revision of the manuscript.

Without repeating all the points listed below, among the more fundamental points are the following:

- The referees list a series of issues regarding the analyses resulting in the 'identification of key-driver genes'. Importantly, they point out that the integration of tissue-specific gene expression data with eSNPs from other tissues could result in the identification of false positives, that updated information on PPIs should be used for the analysis and that the parameters used for defining the clusters need to be carefully considered.
 - Additional experimentation is required in order to better validate the novel key driver genes. Referees #2 and #3 offer constructive suggestions in this regard.
 - The manuscript should be carefully re-written and the presented analyses, datasets and methods need to be described in better detail.
-

Reviewer #1:

The authors employed a systems biology approach that integrates multiple types of genomic data sets to infer causal genes for blood pressure (BP) regulation. Specifically, they first identified 83 differentially expressed genes associated with BP (i.e. the top BP signatures gene set) by using gene expression profiling from whole blood collected from 3,679 FHS participants. They also identified six co-expression modules (coEM) significantly associated with BP traits using the same data set. Then, three of those seven gene sets (1 top BP signatures gene set + 6 coEMs) were further selected and defined as "putatively causal BP gene sets," based on overall significance of the ICBP BP GWAS p-values of "eSNPs" associated with genes in the given gene sets. (Note: These eSNPs were also identified from the same gene expression data and then further augmented by adding existing eSNPs from BP-relevant tissues). Lastly, genes from the putatively causal BP gene sets were integrated into gene regulatory networks and protein-protein interaction (PPI) networks, and 671 key drivers (KDs) of BP regulation were then identified. Among those KDs, the authors experimentally validated one of the top KDs, the SH2B3 gene, in a KO mouse model, showing that some of the predicted genes indeed exhibit the differential expression pattern in SH2B3 deficient mice when hypertension conditions were induced by injection of Ang II.

Although the experimentally tested SH2B3 gene seems genuine as further demonstrated in the companion manuscript (Saleh et al., submitted) and the gene expression data set would be a useful resource for the scientific community, I have the following substantial concerns about the presented computational approaches. Overall, I am not convinced that the proposed method can identify key drivers for BP regulation.

1) Given that gene expression profiles are generally tissue-specific and that the expression data set used in this study was derived from whole blood, incorporation of the existing eSNPs from any other tissues (adipose, liver, and brain) would significantly decrease specificity and increase false positive detection. Several studies have already shown that eQTLs are mostly cell-type specific (i.e. Dimas et al., 2009, Science). The same concerns also apply to the regulatory/PPI network analysis. Only data from blood or blood-related cells (i.e. eQTLs from lymphoblastoid cell lines Lappalainen et al., 2013, Nature) would be useful.

2) The identified co-regulated modules are suspicious too. For example, 2396, 2151, and 88 genes are in the putatively causal BP gene sets (turquoise, blue, and lightyellow respectively), which consist of approximately 30~40% of genes expressed in a given cell type, and they seem to be very loosely correlated, if at all. Also, GO terms associated with these gene sets (especially turquoise and blue coEMs) are general and less informative. What are the average correlation coefficients? Did authors carry out any cluster stability/robustness analysis? Authors should be more careful with the parameters that determine the clusters.

3) Most of the predicted key driver genes also seem unlikely. 530 out of 671 are primarily evidenced by PPI, which is well known to suffer from high false positive rates (Cusick et al., 2005, Hum Mol Gen). Combined with large clusters, the PPI network seems to play a major role in providing false predictions of KDs.

Minor points:

1) Reference for PPI should be the latest one "Human Protein Reference Database--2009 update" (Prasad et al., 2009, Nucleic Acids Res)

2) The term "putatively causal BP gene sets" is misleading. Changes in gene expression associated with genotypes can still be reactive

Reviewer #2:

Summary

The authors have generated gene expression data (using Affymetrix Human Exon arrays ST 1.0)

from whole blood RNA isolated from 3679 non-hypertensive individuals of the Framingham Heart Study (FHS) and are integrated this with published data from GWAS of blood pressure, other GGE studies and the text-mined driven PPI from HPRD to i) identify signature genes for BP, ii)BP-coEMs and iii) eQTLs.

Next, the authors examine which of the 27 CoEMs that also have eQTLs that either are risk enriched ($P=0.05$) or are in LD with known GWA SNPs for BP. The authors conclude that such CoEMs are likely causal for BP. Next Bayesian and PPI networks are used to identify KDs of the coEMs.

They conclude that their integrative approach help to gain insights into novel gene targets in the form of regulatory key driver genes that may help to better understand the molecular underpinnings of BP and possibly suggest novel targets for therapy.

General remarks

Some of the authors are well established in the systems analysis field and the conceptual steps (Fig.1) are largely agreeable and justified. I have, however, some major concerns relating to some of these steps that will need careful consideration before the manuscript is sufficiently strong in my meaning. In addition, I think the validation using independent biological experiments need to be expanded in analyzing some additional KD genes that the authors claim are key regulators for BP. Preferably not one of the key drivers already supported by GWAS or other earlier studies but a driver that principally was found as a result of the current study.

This said, I think the enriched causal networks for BP (if supported by some more biological validation to further reassure their biological relevance) advance our understanding of BP compared to previous knowledge particular how some genes previously related to PB are connected in networks and KDs. Clearly, the manuscript is of interest to a broad scientific audience and particular to those who are focusing on understanding the molecular mechanisms and pathways/networks for established SNP found in GWAS for BP.

Major points

1. Use of adjusted (for 3 major blood cell types) vs. unadjusted whole blood RNA expression data. This part is confusing - I think I am to understand that adjusted values were used for all results forwarded in the paper? It is however not clearly stated what data was used for the PB-related set, eQTLs, coEMs etc. Please state what is used for what in M&Ms. It is also unclear if coEMs related to BP were more overlapping (Sup. Figure 1 and 2) than those that were unrelated - or if for example, only adjusted coEMs were found to related to BP. As stated below, the author can be much clearer on this important point. Clarifying this might support the results as meaningful for BP in my meaning (see also minor comment on eQTLs below).
2. It is unclear to me how the BP-signature set enriches the analysis in a separate way from the coEMs. The coEMs are also based on co-expression generating a gene cluster from which an eigengene is calculated and associated with BP. To me it is no surprise the BP-signature set genes also are part of the coEMs related to BP. The whole point with systems analysis is to avoid single gene analysis (like this) and instead focus on multiple genes in networks/clusters. To me it is either not surprising most of the BP-signature set genes end up in reactive coEMs as opposed to the two causal ones (Table 2). Either exclude them from the analysis altogether or the authors need to explain for me how they believe this set contributes differently from the coEMs. Of note, it would have been different if there also had been a gene expression dataset from hypertensive patients that could have been used to select the most BP-related genes prior to the coEM inference as opposed to use all genes as now. As it seems to me, this signature set does not add credibility to the top KDs.
3. Additional validation experiment as alluded to above. Stating that 2 of 3 genes in a GWA indicated BP gene (SH2B3) mouse KO model are differentially expressed in whole blood RNA upon inducing hypertension just does not cut it. Are there not other available mouse KO models for some of the BP-unrelated KDs of the multi-tissue network in which hypertension can be induced in the same way and compared with controls as of the extent of HT/DE? If not, how about inducing HT in wild type mice and isolate whole blood RNA before and after and examine specifically the KDs (and random background group of genes) to determine if the KDs are the most responsive? Or further validate KDs previously unrelated to BP in some other convincing way that the authors can think of.

Minor points

-It is (to say the least) annoying that the authors almost seem to purposely avoid clearly stating the source of different datasets. Although a minor point (since it can be found after some searching in Sup material etc.) this needs to be better stated making the ms much easier to follow. For example, it should be mentioned in the Abstract (3rd line; whole-blood-RNA (replace "gene") expression profiles), Introduction last Paragraph ".....and transcriptomic data from whole blood (and here also state) adjusted for the three major white blood cell types isolated from 3679...."

- Is the gene expression data from whole blood newly generated or has it been published before?

- Generally the authors repeat describing the flow of the analytical steps too often- it is sufficient to describe this in the Introduction (as now) and be more straight forward in the Result section and particular not repeat the steps again in the Discussion but instead discuss possible pros and cons with chosen methods, alternative approaches that could have been chosen and why this was not felt appropriate.

- 27 coEMs were constructed when seeking relation to PB using eigengenes for the coEM - where these also adjusted for multiple testing?

-The Chocolate coEM is enriched for immune cell mediated cytotoxicity but later dismissed as "reactive" - still the SH2B3 key driver subnetwork claimed to drive BP also was also found to be enriched for T-cell activation - seems to me that many genes here could be in common in both sets - how do you explain this?

-The section on how KDs were identified from BNs and PPIs in the results section is hard to follow. The sentence "At KD-enrichment $p < 0.05$, we identified 671 KDs" is not clarifying. The main principles for calling the KDs need to be better explained.

- early in the section named "Inferring BP gene regulatory subnetworks driven by top key drivers" the authors refer to Table 4 I believe they mean Table 5.

- As already stated, the Discussion is too repetitive and should instead be used to address cons and pros as for example responses to the Major Concerns. Also, in the end of the third paragraph it is stated "Besides these top KDs, many of the other KDs (presumably among the 671) may also be interesting. For example, WNK1.....). I think the authors should at least can give 5 such examples to better match their first statement of "many".

Materials and Methods

- Please in a new Supplementary Figure show distribution of particular DBP since 80 to 90 is considered pre-hypertensive so it would be nice to see how many "normotensive" there actually are.

- eSNPs I think in a table to declare, which of the eQTL sets that were most informative for each coEM related to BP particular for those find to be causal (Light yellow, Turquoise and Blue)- this can perhaps be used to support the results and that the whole-blood RNA data is relevant for BP if many eQTLs were for the coEM derive from there.

- There are some spelling and formulation issues in the Supplementary Note.

Reviewer #3:

This manuscript presented an integrative analysis, which was designed to identify the key elements associated with blood pressure regulation. The authors sampled blood samples from a cohort of FHS participants, and identified genes potentially showing differential expression among individuals with hypertension. The following co-expression analysis, combined with investigations from other functional genomic data, identified several putative key elements potentially involved in blood pressure regulation. One candidate gene SH2B3 was then further tested with a mouse model. Compared with previous GWAS studies, this study presented a more comprehensive view to further understand the disease. While overall this study is interesting, the following issues have to be addressed properly.

Major issues -

1. The paper is a bit hard to follow for two reasons. (a). Too little information is provided in the Result section, and there is even no a simple description for the data in which they were generated and analyzed. For example, at the beginning, in the section of identifying the differentially expressed genes, the authors at least need to describe how many cases and how many control subjects in text, not simply directing the audience to a supplementary table. Also it is also important to show how the ethnic backgrounds, age/sex, ect. were controlled for this comparison. Part of this information can be seen from the Methods section, but it is important to describe the data in the result section as well. Most importantly, the authors have made a great effort in profiling gene

expression in a large number of FHS participants, but throughout the Result/Discussion, there is even no a brief mentioning of the data sources. (b). In the analysis part, the authors seemed to have tried many methods on data from different dimensions; however the overall structure was quite loose, just like piling up the data without strongly logic connections. Overall, I would strongly suggest the authors to rewrite this paper to improve its clarity and logic connectivity. Especially, the Result/Discussion section should be substantially re-written to make itself as a complete story.

2. The large fraction of this paper was based on the co-expression analysis, but the authors have to give more reasoning for their motivations. (1). It is unclear what Types of data were used for this co-expression analysis- gene expression in blood across different individuals in the cohort, just cases or just control subjects? (2). The expression data were across different individuals, so the co-expression analysis identified gene modules showing co-expression in some individuals but were not co-expressed in some other individuals. So this co-expression essentially tested gene expression variability across individuals (as opposed to studying tissue specificity in earlier studies involving the same WGCNA: co-expression analysis across different tissue types), where genes within the same module are presumably showing consistent variability for a sub-group of the cohort. If this is the case, for the modules identified in this study, do they show consistent changes across individuals with high blood pressure only, or the effects were just mixed among patients and healthy subjects? If the latter is true, how can we explain this and justify the effects of co-expression analysis on prioritizing clinically important genes.

3. SSEA analysis identified genes in the co-expression module with significant eSNPs, and how is this expected by chance? e.g. "There are 13 genes in the putatively causal BP gene sets (Turquoise, Blue, and Lightyellow coEMs) whose eSNPs reached $p < 5e-8$ in the ICBP BP GWAS" - do they show an overall enrichment? The authors need to make a logic connection between the above co-expression analysis down to the SSEA analysis, and justify why they need to perform SSEA in the identified co-expression modules.

4. In the section of "Identification of key drivers", the authors need to disclose more details on their analysis, not simply presenting the P-values - how the P-values, how you identified "Key driver genes", and why they are the "KEY" driver? As this analysis was only focused on the blood pressure phenotype, I cannot see the point using tissue-specificity Bayesian networks. First of all, this dataset was just a mixture of data from different sources (including a mouse dataset, Methods section), and the substantial heterogeneity will hurt, rather than, improve the robustness of your study. At least the authors need to justify this before using the dataset. Second, the BN data were not simply biological measurements, but were from predictions, and their confidence needs to be further calibrated. Most importantly, the authors are studying blood pressure, why using the BN data based on liver, adipose and kidney as well? For the HPRD dataset, the latest update was Apr 13, 2010. The dataset was obsolete, and the authors are encouraged to replicate this study on BioGrid interactions.

5. For the gene candidate in SH2B3 in this study, it was great that the authors were able to generate a mouse model to test its effects, but disappointingly, the effects were only tested on 3 genes, so it is really hard to say if this was a merely a chance finding, or indeed support the authors' hypothesis. It is thus strongly recommended that the authors perform a transcriptome study for this mouse model, and then examine how many genes in their prediction also showed dys-regulation in the mouse model. Moreover, what's the phenotype of the SH2B3 knockout mouse? For MGI, the mouse phenotype for SH2B3 knockout is "Mice homozygous for a knock-out allele exhibit severe perturbations in hematopoiesis", how this would be connected with the gene's function in blood pressure regulation?

Minor issues -

1. Statistical robustness- for the WGCNA co-expression analysis, the authors need to perform the modular division for at least 100 times, and the estimate the frequency of each gene that is assigned with a particular module.

2. The sentence - "The top BP signatures gene set and the Chocolate coEM did not show enrichment for BP GWAS eSNPs, suggesting that the gene regulatory structure represented by these gene sets is likely reactive (i.e., downstream) rather than causal for BP." The authors need to back up this claim, and it is possible that they have nothing to do with the GWAS eSNPs.

Overall this manuscript presents an interesting study, but substantial revision is needed to improve its writing clarity and technical solidity.

1st Revision - authors' response

02 December 2014

The authors appreciate the insightful and constructive comments from the editor and the three reviewers. We have revised the manuscript to address each of the comments. We believe that the revised manuscript has improved as a result of the numerous suggestions we received. Major changes are highlighted in the revised manuscript. In the paragraphs that follow, the reviewers' comments are highlighted in bold text and our replies in normal font.

Editor's comments:

1. The referees list a series of issues regarding the analyses resulting in the 'identification of key-driver genes'. Importantly, they point out that the integration of tissue-specific gene expression data with eSNPs from other tissues could result in the identification of false positives, that updated information on PPIs should be used for the analysis and that the parameters used for defining the clusters need to be carefully considered.

Reply: We are grateful for the suggestions and have conducted additional analyses to address each of these three issues, as detailed below. We are pleased to report that our main findings and conclusions did not change after refining the analyses based on the reviewer's suggestions.

1a. The integration of tissue-specific gene expression data with eSNPs from other tissues could result in the identification of false positives.

Our previous rationale for including eSNPs from multiple tissues such as kidney, liver, and adipose tissue, was to increase the coverage of mapping between functional SNPs and their target genes due to the limited number of blood eSNPs available at the time of the analysis. As more blood eSNPs have been generated recently (Battle et al, 2014; Joehanes R. et al, 2013; Lappalainen et al, 2013; Westra et al, 2013; Wright et al, 2014), it is now feasible to use blood-derived eQTLs without other eQTLs from tissues, as suggested by the reviewer. As recommended by the reviewer, we have repeated our analysis using only eSNPs from blood and updated the SNP set enrichment analysis results in Table 3 in the revised manuscript. The Turquoise, Blue and Red coexpression modules (coEMs) that were significant in our previous multi-tissue analysis remained significant in the new analysis; the BP gene expression signature remained non-significant. In addition, the use of blood-only eSNPs identified the Chocolate module as significant.

Main Text Table 3: SNP set enrichment analysis of BP coexpression modules and BP signature gene set.

MODULE	DBP-GWAS		SBP-GWAS	
	KS test pval	Fisher Test pval	KS test pval	Fisher Test pval
BP signature	0.20	1	0.98	1
Turquoise	1.8e-28	3.0e-39	2.8e-45	7.8e-115
Blue	1.3e-8	3.4e-15	1.4e-44	7.0e-54
Red	2.2e-15	6.7e-19	8.0e-5	1.7e-17
Purple	1	1	0.65	0.58
Lightyellow	0.12	1	1.6e-3	1
Chocolate	0.07	1	2.3e-14	5.0e-5

Highlighted p values pass Bonferroni-correction for multiple testing, at Bonferroni-corrected $p < 0.05$.

We also made corresponding changes to the downstream analyses by focusing on the four putatively causal coEMs based on blood eSNP analysis (i.e., Turquoise, Blue, Red and Chocolate coEMs) and highlighted the changes in the revised manuscript.

1b. Updated information on PPIs should be used for the analysis

Indeed, there are several databases that curate PPI information, including HPRD (Keshava Prasad et al, 2009), BioGrid (Chatr-Aryamontri et al, 2013), BIND (Isserlin et al, 2011), DIP (Cotter et al, 2004), and IntAct (Kerrien et al, 2011). Although HPRD is one of the older databases and has not been updated since 2010, it has been widely used by the research community and many findings

based on HPRD have been validated (Graham & Graeber, 2014; Network, 2011; Wu et al, 2014). In addition, the reliability and reproducibility of the PPIs collected in HPRD appear to be superior to those collected in the other databases, as shown in our comparison analysis in Table 1 of this response letter (below). HPRD demonstrated the best overall overlap ratios with the other databases, with the highest ratio observed being 15% between HPRD and BioGrid. Therefore, we conclude that the PPIs in these two databases appear to be more reproducible.

Response Letter Table 1: The overlap of PPIs among different PPI databases

	PPIs	Latest Update	HPRD	BioGrid	BIND	DIP	IntAct
HPRD	65227	Prasad et al. NAR, 2009	--	0.15	0.06	0.02	0.07
BioGRID	124035	Chatr-Aryamontri, et al. NAR, 2013	0.15	--	0.03	0.01	0.08
BIND	19352	Isserlin, R et al. Database, 2011	0.06	0.03	--	0.02	0.02
DIP	2925	Salwinski et al. NAR 2004	0.02	0.01	0.02	--	0.01
IntAct	37629	Kerrien et al. NAR 2012	0.07	0.08	0.02	0.01	--

The numbers in column 4 to 8 indicated the overlap ratio between two databases (db) in the respective column and row. The overlap ratio is defined as $(PPI_{db1} \cap PPI_{db2}) / (PPI_{db1} \cup PPI_{db2})$.

The numbers in column 4 to 8 indicated the overlap ratio between two databases (db) in the respective column and row. The overlap ratio is defined as $(PPI_{db1} \cap PPI_{db2}) / (PPI_{db1} \cup PPI_{db2})$.

To evaluate the reliability of key drivers (KDs) identified in HPRD, we also used BioGrid to identify KDs. We found that 36% of KDs identified in HPRD replicated in BioGrid for the 4 putatively causal BP coEMs, and 50% of KDs in the Turquoise module replicated. The details of the replication of KDs in the two PPI networks were added to the revised manuscript.

1c. The parameters used for defining the clusters need to be carefully considered.

The parameters that we used to construct the gene coexpression network were based on the fitting of the coexpression network in a scale-free topological structure, as recommended by the authors of WGCNA (Langfelder & Horvath, 2008; Zhang & Horvath, 2005). In our study, the fitting index of a scale-free network is 0.99 (the perfect fitting index is 1). The gene clustering tree (dendrogram) of each coEM was obtained from an average linkage hierarchical clustering of the adjacency-based dissimilarity metric. The dynamic tree-cut algorithm was then used to choose the cut-off for module identification (Langfelder et al, 2008).

To confirm the reliability of the coexpression modules (coEMs) or clusters identified, we conducted a robustness test of the coexpression network analysis using a re-sampling strategy. We resampled 80% of the samples ten times, and built a coexpression network from each re-sampled dataset to identify coEMs. To assess the conservation / stability score for a given module M in the original network, we used Fisher's exact test to evaluate the consistency between M and its corresponding modules M-1, M-2, ..., M-10 in the bootstrapped network to derive the overlap p values P(M-1), P(M-2), ..., P(M-10). The maximum (note NOT the minimum, to be conservative) overlap p-value was used as the final conservation /stability p value for each module. The conservation p values for the 6 BP coEMs were all less than 1e-16, supporting the robustness of the coEMs identified in our study.

2. Additional experimentation is required in order to better validate the novel key driver genes. Referees #2 and #3 offer constructive suggestions in this regard.

Reply: We have considered the reviewers' suggestions and conducted a costly but invaluable experiment to validate the *SH2B3* KD and its related subnetworks by examining the transcriptome-level changes in the whole blood of *Sh2b3* knockout mice compared with wide-type (WT) mice (n=4 in each group) using RNA-Seq. We identified 2240 differentially expressed genes that were affected by the knockout of *Sh2b3*, and found highly significant overlap between the genes affected by *Sh2b3* knockout in the mouse model and our predicted *SH2B3* subnetworks, i.e., the *SH2B3* derived genetic subnetwork and the *SH2B3* derived PPI subnetwork (Figure 3 and Table 6 in the revised manuscript).

Main Text Figure 3: *SH2B3* related genetic and protein-protein interaction subnetworks. A) *rs3184504*, a missense SNP, is located in the third exon of *SH2B3*; B) *SH2B3* genetic subnetwork. *rs3184504* is associated with 19 genes in a cis or trans manner based on analysis of eQTLs; C) *SH2B3* protein-protein interaction (PPI) subnetwork. *SH2B3* is depicted as a rectangular node. Green nodes indicate differentially expressed BP genes at Bonferroni corrected $p < 0.05$ in the Framingham Heart Study (BP Top Sig); turquoise nodes indicate genes in the BP causal coEMs; yellow nodes indicate genes that are in both the BP Top Sig set and the BP causal coEMs. The nodes marked with a red border indicate differentially expressed genes between wide-type (WT) and *Sh2b3*^{-/-} mice.

Main Text Table 6: Summary of the overlap between gene signatures of *Sh2b3*^{-/-} mice and the predicted *SH2B3* subnetworks.

SH2B3 subnetwork	Number of genes in the subnetwork	Number of Overlapping genes	Fold Enrichment	p value
Genetic subnetwork	19	8	2.5	1.2e-5
PPI subnetwork	362	78	1.3	2.2e-14

These additional validation studies along with our related study, which shows that *SH2B3* plays a key role in the development of hypertension in mice (Saleh M. et al., under revision in *The Journal of Clinical Investigation*), strongly support the validity of the BP gene networks and the KD *SH2B3*. The detailed experimental methods, new experimental results (under the sub-section “*Validation of the SH2B3 subnetworks using a Sh2b3*^{-/-} mouse model”), and discussion of the validation experiment have been added to the revised manuscript.

The reviewers also suggested that we validate other novel KDs in addition to *SH2B3*. Given the months of time and considerable cost to experimentally validate *SH2B3* in a knockout mouse model, it is beyond the scope (and budget) of the current study to validate additional novel KDs. We have emphasized the need to validate additional novel KDs as future directions

3. The manuscript should be carefully re-written and the presented analyses, datasets and methods need to be described in better detail.

Reply: As requested, we have extensively revised our manuscript to provide sufficient rationale and methodological details for each analysis and to make the paper easier to follow.

**Reviewer #1:
Major points**

1.1. Given that gene expression profiles are generally tissue-specific and that the expression data set used in this study was derived from whole blood, incorporation of the existing eSNPs from any other tissues (adipose, liver, and brain) would significantly decrease specificity and increase false positive detection. Several studies have already shown that eQTLs are mostly cell-type specific (i.e. Dimas et al., 2009, Science). The same concerns also apply to the regulatory/PPI network analysis. Only data from blood or blood-related cells (i.e. eQTLs from lymphoblastoid cell lines Lappalainen et al., 2013, Nature) would be useful.

Reply: As suggested by the reviewer, we have updated our analysis using only eQTLs derived from blood or lymphoblastoid cell lines. We are pleased to report that our main findings and conclusions did not change appreciably, as detailed in our response to the Editor's comment 1a on page 1 of this response letter.

1.2. The identified co-regulated modules are suspicious too. For example, 2396, 2151, and 88 genes are in the putatively causal BP gene sets (turquoise, blue, and lightyellow respectively), which consist of approximately 30~40% of genes expressed in a given cell type, and they seem to be very loosely correlated, if at all. Also, GO terms associated with these gene sets (especially turquoise and blue coEMs) are general and less informative. What are the average correlation coefficients? Did authors carry out any cluster stability/robustness analysis? Authors should be more careful with the parameters that determine the clusters.

Reply: As detailed in our response to the Editor's comment 1c on page 2-3 of this response letter, the parameters that we used to construct the gene coexpression network and determine the clusters (coexpression network modules [coEMs]) were selected based on the fitting of the resulting coexpression network in a scale-free topological structure and applying dynamic tree-cut algorithm, a requirement for WGCNA (Langfelder & Horvath, 2008; Zhang & Horvath, 2005). In our study, the fitting index of a scale-free network is 0.99 (the perfect fitting index is 1), indicating fulfillment of the scale-free topology. As suggested, we also conducted a robustness test of the coexpression network analysis using a re-sampling strategy (detailed on page 2-3 under Editor's comment 1c in this response letter). Our results support the robustness of the coEMs identified in our study.

Although the sizes of several coEMs appear to be large, they are in agreement with those observed in many previous coexpression networks constructed from diverse tissue types (Horvath et al, 2012a; Yang et al, 2010; Zhang et al, 2013). The largest modules typically contain thousands of genes, consistent with our finding. The average absolute correlation coefficients among genes in the BP coEMs are 0.29 (Turquoise), 0.12 (Blue), 0.10 (Red), 0.12 (Purple), 0.25 (Lightyellow) and 0.30 (Chocolate), whereas the average absolute correlation coefficient of the non-module genes is 0.04. These results suggest a tight correlation among genes within each BP coEM. Even though there are 2394 genes in the Turquoise module, the correlation strength among genes in this module is actually very strong. Similarly, although the GO term annotations for certain coEMs are general, these general terms do not undermine the co-regulation structure (defined by data) and the biological importance and relevance of these coEMs.

1.3. Most of the predicted key driver genes also seem unlikely. 530 out of 671 are primarily evidenced by PPI, which is well known to suffer from high false positive rates (Cusick et al., 2005, Hum Mol Gen). Combined with large clusters, the PPI network seems to play a major role in providing false predictions of KDs.

Reply: We agree with the reviewer that the technical limitations of PPI measurements may lead to inaccurate PPI predictions and false positives. However, these limitations are difficult to avoid and many studies have demonstrated the merit of PPI networks for identification and prioritization of

candidate disease genes (Barrenas et al, 2012; Graham & Graeber, 2014; Han et al, 2013; Jia et al, 2012; Ma et al, 2012; Network, 2011; Wu et al, 2014).

In our study, we adopted several strategies to reduce false positives in the reported key drivers (KDs). First, we chose the human PPI databases carefully. To our knowledge, current PPI databases are based on either 1) manual curation of the scientific literature of PPIs with experimental evidence, 2) automated text mining of articles, or 3) computational predictions. PPIs from 2) and 3) yield much higher false positives than from 1). Therefore, we used HPRD (Keshava Prasad et al, 2009) where PPIs were manually curated from the literature by biologists and only PPIs with experimental evidence were included. As detailed in our response to Editor's comment 1b on page 2, the reliability and reproducibility of the PPIs collected in HPRD appear to be comparable to those collected in BioGrid (Chatr-Aryamontri et al, 2013) and superior to those in other major databases such as BIND (Isserlin et al, 2011), DIP (Cotter et al, 2004), and IntAct (Kerrien et al, 2011). Second, to evaluate the reliability of KDs, we also used BioGrid to identify KDs in the revised manuscript. We found that 35% of KDs identified in HPRD were replicated in BioGrid for the four putatively causal BP coEMs, and 50% of KDs in the Turquoise module were replicated. Third, we adopted multiple additional criteria, including BP GWAS results and differential expression values, to further rank KDs. We also validated one of the top KDs, *SH2B3* and its derived subnetwork by examining the transcriptomic changes between wide-type (WT) and *Sh2b3* knockout mice using RNA sequencing, as detailed in the revised manuscript and the response to Editor's comment 2.

Minor points:

1.4. Reference for PPI should be the latest one "Human Protein Reference Database--2009 update" (Prasad et al., 2009, Nucleic Acids Res)

Reply: We corrected the citation of HPRD as suggested by the reviewer.

1.5. The term "putatively causal BP gene sets" is misleading. Changes in gene expression associated with genotypes can still be reactive

Reply: Although we agree with the reviewer that exceptions may exist, there are numerous studies supporting the notion that genes whose expression levels change as a result of genetic perturbations (i.e., genes with eQTLs) are more enriched for disease causal genes (Chen et al, 2008; Civelek & Lusis, 2014; Li et al, 2014; Schadt et al, 2005; Schadt et al, 2008; Yang et al, 2009; Zhang et al, 2013). We agree with the reviewer that any strong claims of causality are unfounded without experimental validation. We used the phrase "*putatively causal*" throughout the manuscript to indicate they are potentially causal based on statistical inference but the causal nature is in need of future experimental validation. In the revised manuscript page 9, we clarified this point as follows:

"A BP gene set showing significance in SSEA is referred to as "putatively causal", because it is supported by orthogonal genetic evidence that is unlikely to be confounded by non-genetic factors. The term "putatively causal" also implies that further experimental validation is needed to prove causality."

Reviewer #2:**Major points**

2.1. Use of adjusted (for 3 major blood cell types) vs. unadjusted whole blood RNA expression data. This part is confusing - I think I am to understand that adjusted values were used for all results forwarded in the paper? It is however not clearly stated what data was used for the BP-related set, eQTLs, coEMs etc. Please state what is used for what in M&Ms.

Reply: We regret the confusion. In the revised manuscript, we report the results from both cell type adjusted and unadjusted data in the main text. As suggested by the reviewer, we have revised the manuscript (see “Materials and Methods” page 18) to clearly state what data was used for which analysis, as following:

“To further evaluate how differences in cell type proportions affect the BP-associated genes at the level of single genes and the coexpression modules identified in this study, we conducted our overall analysis both with and without accounting for cell type effects to capture both cell-type dependent and independent BP-associated genes and processes. We report both results but focus our discussions on those from the adjusted analyses.”

The eQTLs identified using our gene expression data were adjusted for cell types. The whole list of eQTLs will be reported in a separate publication (Joehanes R, PhD, submitted, 2014) as described in the “Materials and Methods” section (page 20) as following:

“For FHS eSNP identification, a pedigree-based linear mixed model was used to determine the association between each gene expression value and the imputed SNP genotypes by adjusting for age, sex, technical covariates, cell types and familial relatedness.”

2.2. It is also unclear if coEMs related to BP were more overlapping (Sup. Figure 1 and 2) than those that were unrelated - or if for example, only adjusted coEMs were found to related to BP. As stated below, the author can be much clearer on this important point. Clarifying this might support the results as meaningful for BP in my meaning (see also minor comment on eQTLs below).

Reply: In the revised manuscript, we clarify the similarities and differences between the results from the adjusted and unadjusted analyses. In summary, we identified 6 BP coEMs using the cell type adjusted data and 7 BP coEMs using the unadjusted data (Supplementary Figure S2, which previously was Supplementary Figure S1). Of these, 3 BP coEMs from the adjusted analysis significantly overlapped with 5 BP coEMs from the unadjusted analysis; 3 and 2 coEMs were unique to the adjusted and unadjusted analysis, respectively. These comparisons supported the presence of both shared and adjustment-specific signals relevant to BP as well as the merit of conducting and reporting both types of analysis. We added a subsection in the Results section (page 6) as follows:

“Influence of blood cell types on gene expression and BP association

As mRNA expression levels might be influenced by differences in the proportions of different cell types in whole blood, we assessed the correlations between mRNAs and three major cell type proportions. We found that approximately 42% of genes were significantly correlated with cell type proportions at Bonferroni corrected $p < 0.05$ (Supplementary Table S1), suggesting a major impact of blood cell types on gene expression. Although results from both cell type adjusted and unadjusted analyses could be biologically relevant (the adjusted analysis may reflect cell type independent signals and the unadjusted analysis may represent cell-type dependent signals), we report both sets of results but focus our discussion on the adjusted analysis to simplify results interpretation. We also report the similarities and differences between the two analyses.”

In the revised manuscript of Results section (page 7-8), we have clarified the main similarities and differences between the coexpression network results from the adjusted and unadjusted analyses as follows:

“Construction of coexpression networks and identification of BP associated gene coexpression modules

....

We also constructed coexpression networks using the data that were unadjusted for cell types (Supplementary Fig S2). We identified 32 coEMs, of which the eigengenes of 7 coEMs (Green,

Greenyellow, Cyan, Magenta, Tan, Midnightblue, and Lightgreen modules) showed significant correlation with SBP or DBP at $p < 0.05$. The Purple and Chocolate coEMs from the cell type adjusted network are conserved with the Green and Lightgreen coEMs from the cell type unadjusted results, respectively. The Turquoise coEM from the adjusted analysis split to three modules in the unadjusted analysis. Other coEMs were unique to the adjusted or unadjusted analysis (Supplementary Fig S3). ”

The details of the comparison of GO enrichment analysis results and SNP set enrichment analysis (SSEA) results between the adjusted and unadjusted coEMs have been added to the revised manuscript (pages 8-10).

2.3. It is unclear to me how the BP-signature set enriches the analysis in a separate way from the coEMs. The coEMs are also based on co-expression generating a gene cluster from which an eigengene is calculated and associated with BP. To me it is no surprise the BP-signature set genes also are part of the coEMs related to BP. The whole point with systems analysis is to avoid single gene analysis (like this) and instead focus on multiple genes in networks/clusters. To me it is either not surprising most of the BP-signature set genes end up in reactive coEMs as opposed to the two causal ones (Table 2). Either exclude them from the analysis altogether or the authors need to explain for me how they believe this set contributes differently from the coEMs. Of note, it would have been different if there also had been a gene expression dataset from hypertensive patients that could have been used to select the most BP-related genes prior to the coEM inference as opposed to use all genes as now. As it seems to me, this signature set does not add credibility to the top KDs.

Reply: We agree with the reviewer that it is important that we focus on network analysis in our systems study. The value of briefly including the differential gene expression signature analysis in our study is to complement our systems biology analysis. As the reviewer has pointed out, most of the traditional studies focus on identifying differential gene expression signatures of diseases using single-gene based analysis; however, how much mechanistic insight these differential expression results provide is a matter of debate. By comparing the single-gene based signature gene sets with the multi-gene focused coEMs, our results support the notion that coEMs capture more BP-related genes and processes than differential expression analysis. This conclusion would not have been possible to reach if the BP signature analysis was not included in the manuscript. By the same token, coEMs can capture BP-related gene coexpression patterns that either highly overlap with BP differential expression signals or are distinct from the BP signatures (Figure 2). If we had pre-selected BP signature genes from hypertensive individuals and only focused on these genes to construct coEMs, we would have missed the coEMs that do not overlap with BP signature genes. Therefore, the BP coEM analysis serves as a comprehensive and powerful means to capture BP etiology; including the traditional expression signature analysis helps to support and highlight the value of our network analysis. In addition, the signature analysis by itself carries more straightforward translational value as potential BP biomarkers since their expression levels are correlated with BP at an individual gene level, and will be of interest to the readers and scientific community. In fact, we have been conducting a thorough differential expression analysis across multiple cohort studies to detect reliable biomarkers of blood pressure, which is the focus of a separate paper.

We also agree with the reviewer that many of the BP-signature genes are likely to be reactive rather than causal based on available evidence. For instance, our SNP set enrichment analysis (SSEA) showed a lack of enrichment with low p value BP GWAS SNPs (Table 3) among the BP-signature set, while several coEMs demonstrated significant enrichment. However, this overall behavior of the BP signature genes as a set does not necessarily preclude a small subset of signature genes, such as, *ATP2B1*, one of the BP signature gene having *cis* eQTLs with BP GWAS $p < 5e-8$, from playing a causal and regulatory role. Another piece of evidence supporting this conclusion comes from the observation that the Red and Chocolate coEMs, which significantly overlap with the BP signature genes, demonstrated enrichment for blood eSNPs with low p value associations in BP GWAS. A possible explanation for this is that the coEMs captured both causal and reactive genes that are functionally correlated. We reason that if a putative KD identified based on gene network topology also shows differential expression, the differential expression status could add credibility to the KD in terms of its relevance to BP, which may help to reduce false positive discoveries in the KD analysis.

2.4. Additional validation experiment as alluded to above. Stating that 2 of 3 genes in a GWA indicated BP gene (SH2B3) mouse KO model are differentially expressed in whole blood RNA upon inducing hypertension just does not cut it. Are there not other available mouse KO models for some of the BP-unrelated KDs of the multi-tissue network in which hypertension can be induced in the same way and compared with controls as of the extent of HT/DE? If not, how about inducing HT in wild type mice and isolate whole blood RNA before and after and examine specifically the KDs (and random background group of genes) to determine if the KDs are the most responsive? Or further validate KDs previously unrelated to BP in some other convincing way that the authors can think of.

Reply: We have taken the reviewer's suggestion and carried out further experimental validation of the *Sh2b3* KD and the corresponding network structure. Going beyond the qPCR-based validation of candidate genes in the previous submission, we conducted RNA sequencing to examine the transcriptomic changes in the whole blood transcriptome of WT and *Sh2b3* KO mice (n=4 in each group) to validate the predicted network structure of *SH2B3*. We identified the differentially expressed genes affected by *Sh2b3* KO and found a highly significant overlap between the genes affected by *Sh2b3* KO in the mouse model and our predicted *SH2B3* subnetwork. This validation study strongly supports both *SH2B3* as a KD and the predicted BP gene network.

Although we appreciate the reviewer's suggestion to validate additional KDs, given the additional months of time and thousands of dollars it has taken to experimentally validate both *SH2B3* as a KD and the derived gene subnetwork, it is beyond the scope of the current study (and our budget) to validate additional KDs.

Minor points

2.5. It is (to say the least) annoying that the authors almost seem to purposely avoid clearly stating the source of different datasets. Although a minor point (since it can be found after some searching in Sup material etc.) this needs to be better stated making the ms much easier to follow. For example, it should be mentioned in the Abstract (3rd line; whole-blood-RNA (replace "gene") expression profiles), Introduction last Paragraph ".....and transcriptomic data from whole blood (and here also state) adjusted for the three major white blood cell types isolated from 3679...."

Reply: We thank the reviewer for these constructive comments designed to help us improve the readability and clarity of our manuscript. In addition to making the specific changes suggested by the reviewer, we have extensively revised our manuscript to be accurate and clear with respect to technical and methodological details and to make the paper easier to follow.

2.6. Is the gene expression data from whole blood newly generated or has it been published before?

Reply: The gene expression data for this project were generated as part of a resource for the Systems Approach to Biomarker Research in Cardiovascular Disease (SABRe CVD) initiative for which multiple phenotypes are being studied. All gene expression data have been deposited into dbGaP to allow further analyses by the outside scientific community. We have been conducting collaboration with 5 external cohort studies to detect reliable biomarkers (differentially expressed genes) for BP via meta-analysis of results across all 6 studies (including the FHS), which will be the focus of a separate paper. The resultant BP meta-analysis paper will cite this *Molecular Systems Biology* paper as the FHS gene expression results for BP.

2.7. Generally the authors repeat describing the flow of the analytical steps too often- it is sufficient to describe this in the Introduction (as now) and be more straight forward in the Result section and particular not repeat the steps again in the Discussion but instead discuss possible pros and cons with chosen methods, alternative approaches that could have been chosen and why this was not felt appropriate.

Reply: We appreciate this suggestion. We have reduced the redundancies in the Results and Discussion sections in the revised manuscript. We have also added discussion of the rationale for choosing the analytical approaches and their pros and cons.

2.8. 27 coEMs were constructed when seeking relation to BP using eigengenes for the coEM - where these also adjusted for multiple testing?

Reply: When selecting BP coEMs we did not adjust for multiple tests. Previous coexpression studies demonstrated that interesting findings can be discovered even when a less rigorous p value threshold is used to identify trait-associated coEMs (i.e. not Bonferroni corrected) (Farber, 2010; Leduc et al, 2012; Miller et al, 2008; Miller et al, 2013; Vanderlinden et al, 2013). An additional rationale for using nominal p values to select coEMs is that these coEMs are further integrated with multiple levels of additional information including genetic signals, eQTLs, and Bayesian and PPI gene networks. Integration of multiple levels of data has been proven to reduce false discoveries. Therefore, although the initial selection of coEMs did not involve multiple testing correction and was more inclusive, the additional downstream analyses serve to increase the confidence of the findings.

2.9. The Chocolate coEM is enriched for immune cell mediated cytotoxicity but later dismissed as "reactive" - still the SH2B3 key driver subnetwork claimed to drive BP also was also found to be enriched for T-cell activation - seems to me that many genes here could be in common in both sets - how do you explain this?

Reply: As noted in our reply to Reviewer #1, blood eQTLs are highly relevant to the current study because the FHS transcriptomic data are from whole blood. However, due to the limited availability of blood eQTLs at the time of our previous SSEA analysis, we utilized eQTLs from multiple tissues to increase coverage of functional SNPs and statistical power. The use of multi-tissue eQTLs, instead of blood-only eQTLs, could have masked processes important in blood, such as the "immune cell mediated cytotoxicity" module. In the revised manuscript, we updated our blood eQTL collection by taking into consideration several newly published studies involving very large sample sizes and tested the causality of each BP coEM using the updated eSNPs from blood only. Indeed, we found that the Chocolate module became highly significant in the updated analysis, which reconciles the discrepancy previously observed between the Chocolate coEM and the *SH2B3* subnetwork.

2.10. The section on how KDs were identified from BNs and PPIs in the results section is hard to follow. The sentence "At KD-enrichment $p < 0.05$, we identified 671 KDs" is not clarifying. The main principles for calling the KDs need to be better explained.

Reply: In order to better describe the "key driver analysis", we made multiple changes to the revised manuscript.

1) In the Methods section, we simplified the language used to describe the KD identification algorithm and added a schematic figure (Supplementary Figure S6) to describe "Key driver (KD) analysis" to make it easier to follow. The main text changes are on page 21 as follows:

"Key driver (KD) analysis

For the putatively causal BP gene sets identified by SSEA, we integrated the genes with molecular networks (BNs and PPI networks as described above) to identify key regulators of every BP gene set using KD analysis. The objective of KD analysis was to identify the important genes for a gene set with respect to a given network structure. A KD of a BP causal gene set is defined as a gene whose neighbor genes in the network are significantly enriched for genes in the BP gene set. As illustrated in Supplementary Fig S6, in order to test if gene G in a network (a BN or PPI network) is a KD or not, first, we identified the subnetwork of G by retrieving its directly connected genes (1st-layer neighbor genes), the genes connected by its 1st-layer neighbor genes (2nd-layer neighbor genes), and the genes connected by its 2nd-layer neighbor genes (3rd-layer neighbor genes). Next, we used Fishers' exact test to evaluate if the genes in the subnetwork of G (1st – 3rd-layer neighbor genes of G) show enrichment for genes in the BP causal gene set to derive a KD-enrichment p value. A G that reached a Bonferroni-corrected KD-enrichment $p < 0.05$ was reported as a KD (after correction for the number of genes in a network).

After the identification of KDs for each BP causal coEM in each network (BNs and PPI network), KDs were further ranked by leveraging 1) the BP association p values of the eSNPs in the KD based on results from the ICBP BP GWAS (Ehret et al, 2011); 2) the differential expression association p value for BP from TWAS; and 3) the KD-enrichment p values."

2) In the Results section labeled "Identification of key drivers", we added text to explain the definition of KDs on page 10:

“A KD is defined as a local network hub whose neighbors in its local subnetwork show enrichment for BP genes in the putatively causal gene sets. Due to their central location in the networks, KDs may have broad impact on multiple genes related to BP.”

3) **Supplementary Fig S6: A Schematic Figure of Key Driver Analysis.** In order to test if gene *G* (shown in red) is a KD or not, the subnetwork of *G* is first extracted by retrieving its 1st to 3rd-layer neighbor genes in the network. Subsequently, the enrichment of genes in a given BP gene set (shown in blue) in the subnetwork of *G* is evaluated. *G* is defined as a KD if the subnetwork of *G* is significantly enriched for genes in the tested gene set (evaluated by Fisher’s Exact test; the significant threshold was Bonferroni-corrected for the number of genes in the gene network used).

2.11. Early in the section named "Inferring BP gene regulatory subnetworks driven by top key drivers" the authors refer to Table 4 I believe they mean Table 5.

Reply: We have corrected the error in the revised manuscript. Thank you for picking this up.

2.12. As already stated, the Discussion is to repetitive and should instead be used to address cons and prons as for example responses to the Major Concerns. Also, in the end of the third paragraph it is stated "Besides these top KDs, many of the other KDs (presumably among the 671) may also be interesting. For example, WNK1.....). I think the authors should at least can give 5 such examples to better match their first statement of "many".

Reply: As suggested by the reviewer, we have revised the Discussion section to remove redundancies and discuss the pros/cons of our approaches. We have also added more examples, including *WNK1* on page 15 as follows:

“Many of the KDs that we identified have previously been reported to be involved in BP regulation, including *WNK1* (Choate et al, 2003), *BMP2* (Atkinson et al, 2002; Hamid et al, 2009), *GPXI* (Ardanaz et al, 2010), *TAF1* (Koschinsky et al, 2001), *GYS1* (Groop et al, 1993; Orho-Melander et al, 1999), *CAST* (Kokubo et al, 2006), *IKBKAP* (Kokubo et al, 2006), *MEF2A* (Oishi et al, 2010), and *PPARA* (Bernal-Mizrachi et al, 2003; Bernal-Mizrachi et al, 2007), supporting the validity of our methods.”

2.13. Please in a new Supplementary Figure show distribution of particular DBP since 80 to 90 is considered pre-hypertensive so it would be nice to see how many "normotensive" there actually are.

Reply: As suggested by the reviewer, we have added the distribution of SBP and DBP in the study subjects (n=3679) as Supplementary Figure S1. Hypertension is defined as SBP \geq 140 mm Hg or DBP \geq 90 mm Hg, which represents 11% of the study population. The proportion of pre-hypertensive individuals with SBP 120-139 or DBP 80-89 is 17%. We have also added a summary statement on the data distribution in the revised manuscript on page 6 as follows:

“The average SBP/DBP was 118/74 mm Hg and 11% of participants had hypertension (HTN; defined as SBP \geq 140 mm Hg or DBP \geq 90 mm Hg). Pre-hypertension, defined as a SBP from 120 to 139 mm Hg or DBP from 80 to 89 mm Hg, was present in 17% of our participants.”

Supplementary Fig S1: The distribution of SBP and DBP in the 3679 FHS participants who were not receiving anti-hypertensive treatment. A) Histogram of SBP; B) Histogram of DBP.

2.14. eSNPs I think in a table to declare, which of the eQTL sets that were most informative for each coEM related to BP particular for those find to be causal (Light yellow, Turquoise and Blue) - this can perhaps be used to support the results and that the whole-blood RNA data is relevant for BP if many eQTLs were for the coEM derive from there.

Reply: As suggested by Reviewer #1 and discussed in our responses to the Editor's comment 1a and Reviewer 1 comment 1.1, whole blood derived eQTLs are perhaps the most relevant to the current study. This is because the transcriptomic data used in the current study are derived from whole blood. In the revised manuscript, we repeated the SNP set enrichment analysis (SSEA) using eQTLs derived from blood or blood-derived lymphoblastoid cells (Table 3). Comparison of the new SSEA results based on blood derived eSNPs with our previous results based on multi-tissues eSNPs showed improvement of enrichment signals for the Turquoise, Blue, Red, and Chocolate coEMs, supporting the notion that blood eQTLs are indeed more informative in our study. In addition, among the 299 eSNPs in the four BP causal coEMs that showed association with BP in the ICBP BP GWAS (Ehret et al, 2011) at $p < 1e-5$ from blood, liver, adipose and kidney tissues, 261 were from blood, again supporting the notion that blood-derived eQTLs are the most informative in our study. This is not surprising given that our transcriptomic data are derived from whole blood RNA. We have revised our manuscript to report only results from the blood eQTL analysis.

2.15. There are some spelling and formulation issues in the Supplementary Note.

Reply: We have extensively revised the Supplementary Information, and corrected spelling and formulation issues.

**Reviewer #3:
Major issues**

3.1a. The paper is a bit hard to follow for two reasons. (a). Too little information is provided in the Result section, and there is even no a simple description for the data in which they were generated and analyzed. For example, at the beginning, in the section of identifying the differentially expressed genes, the authors at least need to describe how many cases and how many control subjects in text, not simply directing the audience to a supplementary table. Also it is also important to show how the ethnic backgrounds, age/sex, etc. were controlled for this comparison. Part of this information can be seen from the Methods section, but it is important to describe the data in the result section as well. Most importantly, the authors have made a great effort in profiling gene expression in a large number of FHS participants, but throughout the Result/Discussion, there is even no a brief mentioning of the data sources.

Reply: We appreciate the reviewer's suggestions on how to improve the clarity of the paper. We have added descriptions of the study cohorts, demographics, and data processing into the Results and Discussion sections, and provided more detailed descriptions in the Methods section. In the first paragraph of the Results section on page 6 in the revised manuscript, we describe the effort in the FHS as below:

"The FHS recently launched the Systems Approach to Biomarker Research in Cardiovascular Disease (SABRe CVD) initiative, which seeks to explore and characterize biomarkers and molecular

underpinnings of CVD and its risk factors, including BP. High-throughput gene expression profiles from whole blood derived RNA were generated in 5626 individuals of European ancestry from the FHS offspring (n=2446) and the third generation (n=3180) cohorts. In order to avoid the confounding effects of drug treatment on gene expression levels, the current study was restricted to 3679 participants who were not receiving anti-hypertensive treatment.

The clinical characteristics of the 3679 study participants are summarized in Table 1."

In the Discussion section on page 13, we highlight the FHS efforts dedicated to the genomic data collection as one of the strengths of the study, as follows:

"This study represents a large, single site transcriptome-wide analysis of BP in 3679 individuals who were not receiving antihypertensive drug treatment."

3.1b. In the analysis part, the authors seemed to have tried many methods on data from different dimensions; however the overall structure was quite loose, just like piling up the data without strongly logic connections. Overall, I would strongly suggest the authors to rewrite this paper to improve its clarity and logic connectivity. Especially, the Result/Discussion section should be substantially re-written to make itself as a complete story.

Reply: We appreciate the reviewer's suggestion and have revised our manuscript extensively to make it easier to follow. In the beginning of each subsection of the Results section, we now better describe the rationale for each analysis to build stronger logical connections across the overall study (highlighted in the revised manuscript on page 6-12). In the Discussion section, we also reorganized the paragraphs to improve the flow and provide a stronger story, as follows (page 13):

"This study represents a large, single site transcriptome-wide analysis of BP in 3679 individuals who were not receiving antihypertensive drug treatment. Extending traditional transcriptome-wide analysis that target differentially expressed genes at the individual gene level, we also conducted higher level coexpression network analysis to identify multiple genes demonstrating co-regulatory network structure in the form of coEMs associated with BP. To differentiate causal from reactive roles of the BP-related genes/gene sets from transcriptome-wide analysis, we integrated the differentially expressed genes and the BP coEMs with SNP association results from BP GWAS. To further pinpoint key BP genes and dissect key regulatory mechanisms among the putatively causal BP gene sets, we projected genes within these gene sets onto gene/protein networks and identified key drivers (KDs). These KDs appear to regulate a large number of interacting genes in gene subnetworks and orchestrate multiple biological processes and pathways underlying BP regulation."

3.2a. The large fraction of this paper was based on the co-expression analysis, but the authors have to give more reasoning for their motivations. (a) It is unclear what Types of data were used for this co-expression analysis- gene expression in blood across different individuals in the cohort, just cases or just control subjects?

Reply: We built the gene coexpression network using all 3679 individuals, rather than in normotensive and hypertensive individuals separately. We have added this information to the Methods section (page 18-19) of the revised manuscript as follows:

"Construction of gene coexpression networks and identification of BP-associated coexpression network modules

....

The coexpression network was constructed using the gene expression data from all 3679 individuals who were not on anti-hypertensive treatment, rather than on normotensive and hypertensive individuals separately. Inclusion of all individuals across the full spectrum of BP variability increased our power to capture co-regulated genes associated with BP variability... The relations of coEMs to BP phenotypes were evaluated by correlating the eigengene (first principle component) of each coEM with SBP and DBP across all 3679 participants via Pearson correlation testing; $p < 0.05$ was considered significant."

3.2b. (b) The expression data were across different individuals, so the co-expression analysis identified gene modules showing co-expression in some individuals but were not co-expressed in some other individuals. So this co-expression essentially tested gene expression variability across individuals (as opposed to studying tissue specificity in earlier studies involving the same WGCNA: co-expression analysis across different tissue types), where genes within the

same module are presumably showing consistent variability for a sub-group of the cohort. If this is the case, for the modules identified in this study, do they show consistent changes across individuals with high blood pressure only, or the effects were just mixed among patients and healthy subjects? If the latter is true, how can we explain this and justify the effects of co-expression analysis on prioritizing clinically important genes.

Reply: The advantage of WGCNA is the ability to fully utilize the variability of gene expression across the entire study samples to derive gene modules containing highly coexpressed or coregulated genes. The coexpression modules identified by WGCNA, therefore, represent highly coregulated or coexpressed genes across the entire population, that is, the expression levels of genes within each module vary together between individuals. This type of network demonstrates the overall organization of genes across the genome and the modules represent functionally related genes for a particular trait or disease, but not all modules have to be important for the disease or trait of interest. Similar to our study, a majority of the previous studies utilizing WGCNA focused on single tissue coexpression networks, rather than multi-tissue networks (Chen et al, 2008; Ghazalpour et al, 2006; Horvath et al, 2012b; Yang et al, 2010; Zhang et al, 2013). The relevance of each module to BP in our study was established by correlating the eigengene of each coexpression module with SBP and DBP. In other words, we determined whether a module as a whole co-varies with BP. If the eigengene of a module is significantly correlated with SBP or DBP, this correlation is pertinent to the entire population, not just a subset of the population (e.g., normotensive or hypertensive).

We have revised our manuscript to provide a better explanation and rationale for coexpression network analysis in the Introduction section (page 5) as follows:

“To identify BP coEMs, we first constructed a coexpression network from the gene expression data from all 3679 samples in order to capture coexpression modules containing highly co-regulated genes across all individuals. We then identified BP coEMs whose eigengenes (representing the expression patterns of all genes in each module) demonstrated significant correlations with BP measurements. The advantage of a coexpression network approach is that it provides a contextual framework to determine the relationship between the phenotype and functionally related genes across a population.”

3.3. SSEA analysis identified genes in the co-expression module with significant eSNPs, and how is this expected by chance? e.g. "There are 13 genes in the putatively causal BP gene sets (Turquoise, Blue, and Lightyellow coEMs) whose eSNPs reached $p < 5 \times 10^{-8}$ in the ICBP BP GWAS" - do they show an overall enrichment?

Reply: SSEA employs two statistical tests to evaluate if a gene set shows enrichment for eSNPs demonstrating low p values in BP GWAS (rather than random chance). Several coEMs showed statistically significant overall enrichment for eSNPs with low p values in BP GWAS by both the KS test and Fisher's exact test. The extremely low enrichment p values from SSEA indicate the significance of the overall enrichment of these coEMs vs. what is expected by chance. For instance, the Turquoise coEM showed enrichment p values of 1.8×10^{-28} and 3.0×10^{-39} from the KS-test and Fisher's exact test, respectively in SSEA when blood eSNPs were used. These p values indicate that the random chance to observe such overall enrichment is extremely small. In the sentence cited by the reviewer, genes with eSNPs at $p < 5 \times 10^{-8}$ in the ICBP BP GWAS were listed in Table 3 to highlight the top module genes that contributed to the overall enrichment.

3.4. The authors need to make a logic connection between the above co-expression analyses down to the SSEA analysis, and justify why they need to perform SSEA in the identified co-expression modules.

Reply: In order to better explain why SSEA was performed after identifying the BP coEMs, we have added a paragraph in the SSEA section in the revised manuscript (page 8-9) as follows:

“Inferring causal modules using SNP set enrichment analysis (SSEA)

The BP gene signatures and coEMs identified above could either play a causal role in regulating BP or just be reactive or independent of BP change. To differentiate these relationships, we used SSEA to evaluate whether the BP gene sets demonstrate enrichment for BP GWAS signals. For each BP-correlated gene set, we first retrieved the blood eSNPs that showed association with the blood expression levels of genes in each BP-correlated gene set, and then extracted the BP association p -values of the eSNPs in the ICBP GWAS. Lastly, the overall distribution of the BP association p

values of the eSNPs representing each BP gene set was compared to the null distribution of all blood eSNPs using two statistical tests, the Kolmogorov-Smirnov (KS) test and Fisher's exact test, to test whether the given BP gene set shows significant enrichment for eSNPs with stronger BP associations (see the Materials and Methods). A BP gene set showing significance in SSEA is referred to as "putatively causal", because it is supported by orthogonal genetic evidence that is unlikely to be confounded by non-genetic factors. The term "putatively causal" also implies that further experimental validation is needed to prove causality."

3.5. In the section of "Identification of key drivers", the authors need to disclose more details on their analysis, not simply presenting the P-values - how the P-values, how you identified "Key driver genes", and why they are the "KEY" driver?

Reply: As discussed in our response to Reviewer #2 comment 2.10 on page 11 of this response letter, we have added detailed descriptions and a Supplementary Figure to better illustrate the "key driver analysis" in the revised manuscript.

3.6. As this analysis was only focused on the blood pressure phenotype, I cannot see the point using tissue-specificity Bayesian networks. First of all, this dataset was just a mixture of data from different sources (including a mouse dataset, Methods section), and the substantial heterogeneity will hurt, rather than, improve the robustness of your study. At least the authors need to justify this before using the dataset. Second, the BN data were not simply biological measurements, but were from predictions, and their confidence needs to be further calibrated. Most importantly, the authors are studying blood pressure, why using the BN data based on liver, adipose and kidney as well?

Reply: Our previous rationale for using Bayesian networks from additional tissues such as kidney, adipose, and liver was that these tissues were previously shown to be relevant to blood pressure (Després et al, 1988; Diehl, 2004; Hall, 2003; Sandok & Whisnant, 1974). However, we agree with the reviewer that blood Bayesian networks are more relevant to the current study. The relevance of blood networks was also supported by our observation that a majority of the KDs from Bayesian networks were identified from blood networks but not from networks from other tissues. We have accepted the reviewer's suggestion and removed the Bayesian networks from the other tissues and have revised the manuscript accordingly.

3.7. For the HPRD dataset, the latest update was Apr 13, 2010. The dataset was obsolete, and the authors are encouraged to replicate this study on BioGrid interactions.

Reply: Although the latest update of HPRD was in 2010, we found that the reliability of the PPIs in HPRD appear to be at least comparable to other PPI databases, as detailed in our response to Editor comment 1b on page 2 of this detailed response. As suggested by the reviewer, we repeated the KD analysis using BioGrid PPI interactions and found that 35% of KDs identified in HPRD were replicated in BioGrid for the three putatively causal BP coEMs, and 50% of the KDs in the Turquoise module were replicated. We have included the new results from the BioGrid analysis in the revised manuscript.

3.8. For the gene candidate in SH2B3 in this study, it was great that the authors were able to generate a mouse model to test its effects, but disappointingly, the effects were only tested on 3 genes, so it is really hard to say if this was a merely a chance finding, or indeed support the authors' hypothesis. It is thus strongly recommended that the authors perform a transcriptome study for this mouse model, and then examine how many genes in their prediction also showed dys-regulation in the mouse model.

Reply: As suggested by the reviewer, we examined the transcriptome-level changes in whole blood of 4 wide-type (WT) and 4 *Sh2b3* knockout (KO) mice using RNA-Seq. We are pleased to report that we observed a highly significant overlap between the differentially expressed genes identified in the mouse model and from our *SH2B3* related gene subnetwork. The new validation data, methods, results, and discussion have been added to the revised manuscript, as detailed in our response to the Editor's comment 2 on page 3-4 of this detailed response.

3.9. Moreover, what's the phenotype of the SH2B3 knockout mouse? For MGI, the mouse phenotype for SH2B3 knockout is "Mice homozygous for a knock-out allele exhibit severe perturbations in hematopoiesis", how this would be connected with the gene's function in blood pressure regulation?

Reply: The phenotypes of *SH2B3*^{-/-} mice have been reported in detail in a separate manuscript (Saleh M. et al, under revision at *Journal of Clinical Investigation*). That study did not include the results of RNA-Seq analysis that are reported exclusively for this manuscript. In our revised manuscript, we briefly describe the phenotypic results of *SH2B3* knockout mice on page 12.

“In a related study (Saleh M. et al., in preparation), we found that Sh2b3^{-/-} mice had normal baseline BP but markedly elevated blood pressure in response to a low dose of angiotensin-II (Ang II; 140 ng/kg/min) that did not affect BP in wild type (WT) mice. This suggests a key role of Sh2b3 in BP regulation, and that loss or changes to this gene exacerbate response to hypertensive stimuli.

...

Consistent with our prediction, Saleh et al. also confirmed the exacerbation of inflammation and T cell activation in Sh2b3^{-/-} mice (Saleh M. et al., in preparation).”

Minor issues

3.10. Statistical robustness- for the WGCNA co-expression analysis, the authors need to perform the modular division for at least 100 times, and the estimate the frequency of each gene that is assigned with a particular module.

Reply: As requested, we used a resampling strategy to validate the robustness of the coEMs in our study. Due to the high computational burden, performing network construction 100 times was not feasible. We took an alternative strategy by resampling 80% of samples ten times, and built 10 coexpression networks to identify coEMs. As detailed in our response to the Editor’s comment 1c on page 2 of this detailed response, this robustness analysis supported the reliability of the coexpression modules identified in our study.

3.11. The sentence - "The top BP signatures gene set and the Chocolate coEM did not show enrichment for BP GWAS eSNPs, suggesting that the gene regulatory structure represented by these gene sets is likely reactive (i.e., downstream) rather than causal for BP." The authors need to back up this claim, and it is possible that they have nothing to do with the GWAS eSNPs.

Reply: We agree with the reviewer that this is purely speculation and a gene module that does not show enrichment for GWAS eSNPs is not necessarily “reactive”. We have removed the sentence in the revised manuscript.

3.12. Overall this manuscript presents an interesting study, but substantial revision is needed to improve its writing clarity and technical solidity.

Reply: We appreciate that the reviewer recognizes the novelty of our study. We have extensively revised our manuscript to improve clarity, established the logic connections between sections, and enhance the technical elements.

References

Ardanaz N, Yang X-P, Cifuentes ME, Haurani MJ, Jackson KW, Liao T-D, Carretero OA, Pagano PJ (2010) Lack of glutathione peroxidase 1 accelerates cardiac-specific hypertrophy and dysfunction in angiotensin II hypertension. *Hypertension* **55**: 116-123

Atkinson C, Stewart S, Upton PD, Machado R, Thomson JR, Trembath RC, Morrell NW (2002) Primary pulmonary hypertension is associated with reduced pulmonary vascular expression of type II bone morphogenetic protein receptor. *Circulation* **105**: 1672-1678

Barhouni T, Kasal DA, Li MW, Shbat L, Laurant P, Neves MF, Paradis P, Schiffrin EL (2011) T Regulatory lymphocytes prevent angiotensin ii-induced hypertension and vascular injury. *Hypertension* **57**: 469-476

Barrenas F, Chavali S, Alves AC, Coin L, Jarvelin M-R, Jornsten R, Langston MA, Ramasamy A, Rogers G, Wang H (2012) Highly interconnected genes in disease-specific networks are enriched for disease-associated polymorphisms. *Genome Biol* **13**: R46

- Battle A, Mostafavi S, Zhu X, Potash JB, Weissman MM, McCormick C, Haudenschild CD, Beckman KB, Shi J, Mei R (2014) Characterizing the genetic basis of transcriptome diversity through RNA-sequencing of 922 individuals. *Genome research* **24**: 14-24
- Bernal-Mizrachi C, Weng S, Feng C, Finck BN, Knutsen RH, Leone TC, Coleman T, Mecham RP, Kelly DP, Semenkovich CF (2003) Dexamethasone induction of hypertension and diabetes is PPAR- α dependent in LDL receptor-null mice. *Nature medicine* **9**: 1069-1075
- Bernal-Mizrachi C, Xiaozhong L, Yin L, Knutsen RH, Howard MJ, Arends JJ, DeSantis P, Coleman T, Semenkovich CF (2007) An afferent vagal nerve pathway links hepatic PPAR α activation to glucocorticoid-induced insulin resistance and hypertension. *Cell metabolism* **5**: 91-102
- Chatr-Aryamontri A, Breitkreutz BJ, Heinicke S, Boucher L, Winter A, Stark C, Nixon J, Ramage L, Kolas N, O'Donnell L, Reguly T, Breitkreutz A, Sellam A, Chen D, Chang C, Rust J, Livstone M, Oughtred R, Dolinski K, Tyers M (2013) The BioGRID interaction database: 2013 update. *Nucleic Acids Res* **41**: D816-823
- Chen Y, Zhu J, Lum PY, Yang X, Pinto S, MacNeil DJ, Zhang C, Lamb J, Edwards S, Sieberts SK, Leonardson A, Castellini LW, Wang S, Champy MF, Zhang B, Emilsson V, Doss S, Ghazalpour A, Horvath S, Drake TA et al (2008) Variations in DNA elucidate molecular networks that cause disease. *Nature* **452**: 429-435
- Choate KA, Kahle KT, Wilson FH, Nelson-Williams C, Lifton RP (2003) WNK1, a kinase mutated in inherited hypertension with hyperkalemia, localizes to diverse Cl⁻-transporting epithelia. *Proc Natl Acad Sci U S A* **100**: 663-668
- Civelek M, Lusis AJ (2014) Systems genetics approaches to understand complex traits. *Nat Rev Genet* **15**: 34-48
- Cotter D, Guda P, Fahy E, Subramaniam S (2004) MitoProteome: mitochondrial protein sequence database and annotation system. *Nucleic acids research* **32**: D463-D467
- Després J-P, Tremblay A, Thériault G, Pérusse L, Leblanc C, Bouchard C (1988) Relationships between body fatness, adipose tissue distribution and blood pressure in men and women. *Journal of clinical epidemiology* **41**: 889-897
- Didion SP, Kinzenbaw DA, Schrader LI, Chu Y, Faraci FM (2009) Endogenous interleukin-10 inhibits angiotensin II-induced vascular dysfunction. *Hypertension* **54**: 619-624
- Diehl A (2004) Fatty liver, hypertension, and the metabolic syndrome. *Gut* **53**: 923-924
- Ehret GB, Munroe PB, Rice KM, Bochud M, Johnson AD, Chasman DI, Smith AV, Tobin MD, Verwoert GC, Hwang SJ, Pihur V, Vollenweider P, O'Reilly PF, Amin N, Bragg-Gresham JL, Teumer A, Glazer NL, Launer L, Zhao JH, Aulchenko Y et al (2011) Genetic variants in novel pathways influence blood pressure and cardiovascular disease risk. *Nature* **478**: 103-109
- Farber CR (2010) Identification of a gene module associated with BMD through the integration of network analysis and genome-wide association data. *Journal of Bone and Mineral Research* **25**: 2359-2367
- Ghazalpour A, Doss S, Zhang B, Wang S, Plaisier C, Castellanos R, Brozell A, Schadt EE, Drake TA, Lusis AJ, Horvath S (2006) Integrating genetic and network analysis to characterize genes related to mouse weight. *PLoS Genet* **2**: e130
- Graham NA, Graeber TG (2014) Complexity of metastasis-associated SDF-1 ligand signaling in breast cancer stem cells. *Proceedings of the National Academy of Sciences* **111**: 7503-7504
- Groop LC, Kankuri M, Schalin-Jantti C, Ekstrand A, Nikula-Ijas P, Widen E, Kuusmanen E, Eriksson J, Franssila-Kallunki A, Saloranta C (1993) Association between polymorphism of the

- glycogen synthase gene and non-insulin-dependent diabetes mellitus. *New England Journal of Medicine* **328**: 10-14
- Hall JE (2003) The kidney, hypertension, and obesity. *Hypertension* **41**: 625-633
- Hamid R, Cogan JD, Hedges LK, Austin E, Phillips JA, Newman JH, Loyd JE (2009) Penetrance of Pulmonary Arterial Hypertension Is Modulated by the Expression of Normal BMPR2 Allele. *Hum Mutat* **30**: 649-654
- Han S, Yang B-Z, Kranzler HR, Liu X, Zhao H, Farrer LA, Boerwinkle E, Potash JB, Gelernter J (2013) Integrating GWASs and human protein interaction networks identifies a gene subnetwork underlying alcohol dependence. *The American Journal of Human Genetics* **93**: 1027-1034
- Harrison DG, Guzik TJ, Lob HE, Madhur MS, Marvar PJ, Thabet SR, Vinh A, Weyand CM (2011) Inflammation, immunity, and hypertension. *Hypertension* **57**: 132-140
- Horvath S, Nazmul-Hossain AN, Pollard RP, Kroese FG, Vissink A, Kallenberg CG, Spijkervet FK, Bootsma H, Michie SA, Gorr SU, Peck AB, Cai C, Zhou H, Wong DT (2012a) Systems analysis of primary Sjogren's syndrome pathogenesis in salivary glands identifies shared pathways in human and a mouse model. *Arthritis Res Ther* **14**: R238
- Horvath S, Zhang Y, Langfelder P, Kahn RS, Boks MP, van Eijk K, van den Berg LH, Ophoff RA (2012b) Aging effects on DNA methylation modules in human brain and blood tissue. *Genome Biol* **13**: R97
- Isserlin R, El-Badrawi RA, Bader GD (2011) The biomolecular interaction network database in PSI-MI 2.5. *Database* **2011**: baq037
- Jia P, Wang L, Fanous AH, Pato CN, Edwards TL, Zhao Z, Consortium IS (2012) Network-assisted investigation of combined causal signals from genome-wide association studies in schizophrenia. *PLoS computational biology* **8**: e1002587
- Joehanes R. , Huan T. , C Yao, X Zhang, S Ying, M Feolo, N. Sharapova, T. Przytycka, A. Sturcke, A. A. Schaffer, N. Heard-Costa, P. Liu, R. Wang KA (2013). Genome-wide Expression Quantitative Trait Loci: Results from the NHLBI's SABRe CVD Initiative. *the American Society of Human Genetics (ASHG) conference*; Oct 22-26; Boston Convention Ctr. Boston, MA.
- Kerrien S, Aranda B, Breuza L, Bridge A, Broackes-Carter F, Chen C, Duesbury M, Dumousseau M, Feuermann M, Hinz U (2011) The IntAct molecular interaction database in 2012. *Nucleic acids research*: gkr1088
- Keshava Prasad TS, Goel R, Kandasamy K, Keerthikumar S, Kumar S, Mathivanan S, Telikicherla D, Raju R, Shafreen B, Venugopal A, Balakrishnan L, Marimuthu A, Banerjee S, Somanathan DS, Sebastian A, Rani S, Ray S, Harrys Kishore CJ, Kanth S, Ahmed M et al (2009) Human Protein Reference Database--2009 update. *Nucleic Acids Res* **37**: D767-772
- Kokubo Y, Tomoike H, Tanaka C, Banno M, Okuda T, Inamoto N, Kamide K, Kawano Y, Miyata T (2006) Association of sixty-one non-synonymous polymorphisms in forty-one hypertension candidate genes with blood pressure variation and hypertension. *Hypertens Res* **29**: 611-619
- Koschinsky ML, Boffa MB, Nesheim ME, Zinman B, Hanley AJ, Harris SB, Cao H, Hegele RA (2001) Association of a single nucleotide polymorphism in CPB2 encoding the thrombin-activable fibrinolysis inhibitor (TAF1) with blood pressure. *Clinical Genetics* **60**: 345-349
- Langfelder P, Horvath S (2008) WGCNA: an R package for weighted correlation network analysis. *BMC Bioinformatics* **9**: 559
- Langfelder P, Zhang B, Horvath S (2008) Defining clusters from a hierarchical cluster tree: the Dynamic Tree Cut package for R. *Bioinformatics* **24**: 719-720

Lappalainen T, Sammeth M, Friedländer MR, AC't Hoen P, Monlong J, Rivas MA, González-Porta M, Kurbatova N, Griebel T, Ferreira PG (2013) Transcriptome and genome sequencing uncovers functional variation in humans. *Nature*

Leduc MS, Blair RH, Verdugo RA, Tsaih S-W, Walsh K, Churchill GA, Paigen B (2012) Using bioinformatics and systems genetics to dissect HDL cholesterol levels in an MRL/MpJ x SM/J intercross. *Journal of lipid research: jlr*. M025833

Li Q, Stram A, Chen C, Kar S, Gayther S, Pharoah P, Haiman C, Stranger B, Kraft P, Freedman ML (2014) Expression QTL-based analyses reveal candidate causal genes and loci across five tumor types. *Hum Mol Genet*

Lin C-C, Bradstreet TR, Schwarzkopf EA, Sim J, Carrero JA, Chou C, Cook LE, Egawa T, Taneja R, Murphy TL (2014) Bhlhe40 controls cytokine production by T cells and is essential for pathogenicity in autoimmune neuroinflammation. *Nature communications* **5**

Ma L, Brautbar A, Boerwinkle E, Sing CF, Clark AG, Keinan A (2012) Knowledge-driven analysis identifies a gene-gene interaction affecting high-density lipoprotein cholesterol levels in multi-ethnic populations. *PLoS genetics* **8**: e1002714

Miller JA, Oldham MC, Geschwind DH (2008) A systems level analysis of transcriptional changes in Alzheimer's disease and normal aging. *The Journal of Neuroscience* **28**: 1410-1420

Miller JA, Woltjer RL, Goodenbour JM, Horvath S, Geschwind DH (2013) Genes and pathways underlying regional and cell type changes in Alzheimer's disease. *Genome Med* **5**: 48

Miyazaki K, Miyazaki M, Guo Y, Yamasaki N, Kanno M, Honda Z, Oda H, Kawamoto H, Honda H (2010) The role of the basic helix-loop-helix transcription factor Dec1 in the regulatory T cells. *J Immunol* **185**: 7330-7339

Network CGAR (2011) Integrated genomic analyses of ovarian carcinoma. *Nature* **474**: 609-615

Oishi Y, Manabe I, Imai Y, Hara K, Horikoshi M, Fujiu K, Tanaka T, Aizawa T, Kadowaki T, Nagai R (2010) Regulatory polymorphism in transcription factor KLF5 at the MEF2 element alters the response to angiotensin II and is associated with human hypertension. *The FASEB Journal* **24**: 1780-1788

Orho-Melander M, Almgren P, Kanninen T, Forsblom C, Groop LC (1999) A paired-sibling analysis of the XbaI polymorphism in the muscle glycogen synthase gene. *Diabetologia* **42**: 1138-1145

Sandok BA, Whisnant JP (1974) Hypertension and the brain. *Archives of internal medicine* **133**: 947-954

Schadt EE, Lamb J, Yang X, Zhu J, Edwards S, Guhathakurta D, Sieberts SK, Monks S, Reitman M, Zhang C, Lum PY, Leonardson A, Thieringer R, Metzger JM, Yang L, Castle J, Zhu H, Kash SF, Drake TA, Sachs A et al (2005) An integrative genomics approach to infer causal associations between gene expression and disease. *Nat Genet* **37**: 710-717

Schadt EE, Molony C, Chudin E, Hao K, Yang X, Lum PY, Kasarskis A, Zhang B, Wang S, Suver C, Zhu J, Millstein J, Sieberts S, Lamb J, GuhaThakurta D, Derry J, Storey JD, Avila-Campillo I, Kruger MJ, Johnson JM et al (2008) Mapping the genetic architecture of gene expression in human liver. *PLoS Biol* **6**: e107

Sun H, Lu B, Li RQ, Flavell RA, Taneja R (2001) Defective T cell activation and autoimmune disorder in Stra13-deficient mice. *Nat Immunol* **2**: 1040-1047

Vanderlinden LA, Saba LM, Kechris K, Miles MF, Hoffman PL, Tabakoff B (2013) Whole brain and brain regional coexpression network interactions associated with predisposition to alcohol consumption. *PLoS One* **8**: e68878

Westra H-J, Peters MJ, Esko T, Yaghootkar H, Schurmann C, Kettunen J, Christiansen MW, Fairfax BP, Schramm K, Powell JE (2013) Systematic identification of trans eQTLs as putative drivers of known disease associations. *Nature genetics* **45**: 1238-1243

Wright FA, Sullivan PF, Brooks AI, Zou F, Sun W, Xia K, Madar V, Jansen R, Chung W, Zhou Y-H (2014) Heritability and genomics of gene expression in peripheral blood. *Nature genetics* **46**: 430-437

Wu X, Renuse S, Sahasrabudde NA, Zahari MS, Chaerkady R, Kim M-S, Nirujogi RS, Mohseni M, Kumar P, Raju R (2014) Activation of diverse signalling pathways by oncogenic PIK3CA mutations. *Nature communications* **5**

Yang X, Deignan JL, Qi H, Zhu J, Qian S, Zhong J, Torosyan G, Majid S, Falkard B, Kleinhanz RR, Karlsson J, Castellani LW, Mumick S, Wang K, Xie T, Coon M, Zhang C, Estrada-Smith D, Farber CR, Wang SS et al (2009) Validation of candidate causal genes for obesity that affect shared metabolic pathways and networks. *Nat Genet* **41**: 415-423

Yang X, Zhang B, Molony C, Chudin E, Hao K, Zhu J, Gaedigk A, Suver C, Zhong H, Leeder JS, Guengerich FP, Strom SC, Schuetz E, Rushmore TH, Ulrich RG, Slatter JG, Schadt EE, Kasarskis A, Lum PY (2010) Systematic genetic and genomic analysis of cytochrome P450 enzyme activities in human liver. *Genome research* **20**: 1020-1036

Zhang B, Gaiteri C, Bodea LG, Wang Z, McElwee J, Podtelezhnikov AA, Zhang C, Xie T, Tran L, Dobrin R, Fluder E, Clurman B, Melquist S, Narayanan M, Suver C, Shah H, Mahajan M, Gillis T, Mysore J, MacDonald ME et al (2013) Integrated systems approach identifies genetic nodes and networks in late-onset Alzheimer's disease. *Cell* **153**: 707-720

Zhang B, Horvath S (2005) A general framework for weighted gene co-expression network analysis. *Statistical applications in genetics and molecular biology* **4**

2nd Editorial Decision

13 January 2015

Thank you again for submitting your work to Molecular Systems Biology. We have now heard back from the three referees who were asked to evaluate your manuscript. As you will see below, the referees think that their main concerns have been satisfactorily addressed. They list however a number of remaining issues, which we would ask you to address in a revision of the manuscript.

Moreover we would like to ask you to draw your attention to the following points, related to the reference to data that are reported in two other manuscripts (cited as 'in preparation'):

- We would like to ask you to provide further information regarding the Sh2b3^{-/-} mouse. In particular, we would ask you to describe the targeting strategy for generating the mouse, provide evidence that the strategy has been successful and include evidence that this is indeed a Sh2b3^{-/-} mouse. We understand that you prefer to describe the full characterization of the phenotype in a separate manuscript, however considering the that this mouse model is crucial for the validation of the SH2B3 controlled sub-networks we think that the above information should also be provided in the present work. Moreover, in case the Saleh et al. study has been already accepted, it should be cited in the paper.
- In the Materials and Methods subsection 'Identification and collection of whole blood eSNPs' you mention: "The eSNPs of whole blood used in this study were combined from eSNPs identified from FHS whole blood gene expression (~18,000 genes) and genotype data (~ 8 million SNPs after imputation to the 1000 Genomes reference panel) (Joehanes R, in preparation)....". We would like to ask you to include a more detailed and clear description of how these eSNPs were obtained and

integrated (i.e. it was not so clear to us how the 8 million SNPs from genotype data were used). Moreover, we would like to ask you to clarify whether newly identified eSNPs, if any (and especially those that are essential for the presented analyses), have been made available via deposition to a database.

Reviewer #1:

The authors did address most concerns raised by the reviewers, but I still have a few more questions.

- 1) In Fig 2b/c, the authors only presented the significant coEMs. However, the remaining non-significant modules should also be presented for completeness. It would be useful to add bar plots including all 27 modules' p-values (maybe ordered by the p-values?) as supplementary figures.
- 2) Fisher's exact test is typically used for categorical data. Therefore, when performing SSEA, a GWAS p-value threshold must have been applied, but has not been specified. What is the threshold?
- 3) It seems that Mann-Whitney/Wilcoxon ranked-sum test is more appropriate for SSEA, because the rank-sum test is mostly sensitive to changes in the median while KS-test is sensitive to ANY differences in the two distributions (i.e. shape, spread, or median). Therefore, I suggest that the authors also calculate the rank-sum test p-values.
- 4) To further strengthen the statistical significance of SSEA, the observed p-values should be compared to the distribution of p-values of random gene sets that match the number of genes.
- 5) In Table 5, how many of these top 20 KDs are also detected by the BioGrid PPI database?

Reviewer #2:

The only remaining concern I have is that I do not agree with the authors that the 32 CoEMs shall not be corrected for multiple testing while correlating the eigengene values with hypertension (not necessarily bonferroni (P_values divided with 32) but FDR-adjusted). I think this will become practice for manuscript in which associations are sought for network modules with phenotypes. I agree, however, that the downstream validation now is so convincing that there are little doubt, if any, of the true value of the HT causal co_EMs particular for the co-EM subnetwork surrounding SH2B3.

Reviewer #3:

The manuscript has been substantially improved in this revision, and the newly added analysis and experiments (mouse RNA-seq) have served to better support the conclusions from this study. Overall, this manuscript presents an interesting study using co-expression network to examine genetic factors potentially implicated in BP traits, and also generated a large-scale expression data which will be valuable resource for the community. Several issues remain:

1. Multiple-hypothesis correct was not conducted in many places in the text. For example, the authors identified 27 coEMs, among which 6 were associated with either SBP or DBP and 2 were associated with BP signature gene set. Since the enrichment test was performed across 27 coEMs, the uncorrected P values should be adjusted. In the same line, almost all the GO enrichment P values in this study were uncorrected. This also includes the enrichment test of KD genes.
2. SSEA analysis section: the size of each co-expression module should be considered. For larger modules with more genes, they may have higher chance to contain expression-SNPs/eQTLs or significance GWAS variants. Therefore, the authors need to control for module sizes in their statistical test.
3. The use of "putative causal" throughout the manuscript is too strong. The enrichment at most only suggests association, but not putative causal.
4. The definition of KD genes is unclear. From what described in the text, A KD gene is defined (i) a local network hub (ii) its network neighborhood shows BP gene enrichment. It is thus unclear if a

KD gene itself is associated with blood pressure or shows differential expression in patients. From the definition given, KD genes could have neither, which is more evidenced against than support their roles as "driver". More importantly, for many genes causal for human diseases, they do not have to be hubs with global impacts on the network; instead, disease might arise by simply ablating the peripheral nodes with a highly specified function. Therefore, the authors need to carefully justify why the "driver" genes in BP traits should be network hubs.

5. In the regulatory association network with SH2B3, the authors showed that many known BP-related genes were involved. Are there any enrichment statistics to support this observation?

6. The overall co-expression network was constructed using the entire selected participant in this study, 28% (11%+17%) with hypertension or pre-hypertension. The author should examine whether the sample heterogeneity would affect their conclusions. A recent study using co-expression network to study autism (Voineagu, I. et al. 2011, Nature) constructed the networks separately for cases and controls, and observed modules showing differences between the two groups. The authors might try to divide the samples according to their BP traits and examine the module differences between individuals with and without hypertension traits, which is complementary to the differential expression study. Given almost ~4000 samples were studied in this study, the sub-division will unlikely compromise the statistical power in the comparisons.

Other issues:

1. In many places, the manuscript cited unpublished results and conclusions. Is it a companion paper sent to MSB, or the authors might consider using other alternative sources to support their data.
2. 83 genes showed differential expression between cases and controls, among which 65 and 8 showed positive and negative correlations with BP traits, respectively. The authors should provide the related statistics (distribution of correlations) in the context and supplementary tables to help the audience better understand the relative contribution of each gene to the BP traits.
3. Overall, this paper shows much "technical development" to uncover the loci or gene groups associated with BP traits, but their potential biology remains less discussed. It seems that the authors did not give much attention to the 83 differentially expressed genes, and relied more on the GWAS signals for eSNPs. It will be useful to discuss their disease implications.

2nd Revision - authors' response

10 February 2015

We appreciate the constructive comments from the Editor and the Reviewers. We have revised the manuscript to address each of the comments. Major changes in the main text are highlighted in the revised manuscript. Below are our point-to-point responses, with the reviewers' comments in bold text.

Editor's comments:

1. We would like to ask you to provide further information regarding the *Sh2b3*^{-/-} mouse. In particular, we would ask you to describe the targeting strategy for generating the mouse, provide evidence that the strategy has been successful and include evidence that this is indeed a *Sh2b3*^{-/-} mouse. We understand that you prefer to describe the full characterization of the phenotype in a separate manuscript, however considering the that this mouse model is crucial for the validation of the SH2B3 controlled sub-networks we think that the above information should also be provided in the present work. Moreover, in case the Saleh et al. study has been already accepted, it should be cited in the paper.

Reply: Our related manuscript, which shows that *SH2B3* (also known as *LNK*) plays a key role in the development of hypertension in mice, has been accepted for publication (Saleh M. et al., "Lymphocyte adaptor protein *LNK* deficiency exacerbates hypertension and end-organ inflammation". *The Journal of Clinical Investigation*. 2015, In press). As requested, we cited this paper in the revised manuscript to support our results and also added additional details of the gene targeting strategy as described by Takaki et al. (Takaki et al, 2000). To prove that the mouse we utilized is indeed a *Sh2b3*^{-/-} mouse, we provided RNA-seq evidence in Supplementary Figure S6, which shows the absence of RNA reads of exons 3-8 of *Sh2b3* in the *Sh2b3*^{-/-} mice, in complete agreement with the design of the targeting construct used to produce these mice (Takaki et al, 2000).

Supplementary Figure S6: Screenshot of RNA reads mapped to *Sh2b3* in a WT mouse and a *Sh2b3*^{-/-} mouse. The RNA reads of exon 3-8 of *Sh2b3* are absent in the *Sh2b3*^{-/-} mouse as expected. This picture is drawn by Integrative Genomics Viewer (IGV) (Thorvaldsdóttir et al, 2012).

2. In the Materials and Methods subsection 'Identification and collection of whole blood eSNPs' you mention: "The eSNPs of whole blood used in this study were combined from eSNPs identified from FHS whole blood gene expression (~18,000 genes) and genotype data (~8 million SNPs after imputation to the 1000 Genomes reference panel) (Joehanes R, in preparation)....". We would like to ask you to include a more detailed and clear description of how these eSNPs were obtained and integrated (i.e. it was not so clear to us how the 8 million SNPs from genotype data were used). Moreover, we would like to ask you to clarify whether newly identified eSNPs, if any (and especially those that are essential for the presented analyses), have been made available via deposition to a database.

Reply: Because the eQTLs generated in Framingham data have not been published yet (Joehanes R, et al. under review in *Nature Communications*), we included more details in the methods section to describe the newly identified eQTLs (page 21). The newly generated eQTL results will be released on the NCBI Molecular QTL Browser (<http://preview.ncbi.nlm.nih.gov/gap/eql/bioprocess/>) when the paper is published. The text of our revised manuscript has been modified as follows:

"The FHS blood eQTLs were generated using data from 5257 FHS participants with genome-wide genotype data and gene expression profiling. DNA isolation, and genotyping with the Affymetrix 500K mapping array and the Affymetrix 50K gene-focused MIP array have been described previously (Levy et al, 2009). Imputation of ~36.3 million SNPs in 1000 Genomes Phase 1 SNP data was conducted using MACH (Li et al, 2010). For the eQTL identification, we used the 1000-genome resource imputed SNPs with minor allele frequency (MAF) >0.01 and imputation ratio >0.3, yielding approximately 8 million SNPs for eQTL analysis. A pedigree-based linear mixed model was used to determine the association between each gene expression value and the imputed SNP genotypes by adjusting for age, sex, technical covariates, cell types, and familial relatedness. The cis eSNPs (or eQTLs) were constrained by a 1 megabyte (Mb) window on either side of the transcription start site (TSS). The remaining eSNPs were defined as trans eSNPs. Genomic coordinates were based on NCBI human reference genome build 37/hg19. The Benjamini-Hochberg method (BH) (Benjamini & Hochberg, 1995) was used to calculate false discovery rates (FDR) of cis and trans eQTLs separately."

Reviewer #1:

1.1 In Figure 2b/c, the authors only presented the significant coEMs. However, the remaining non-significant modules should also be presented for completeness. It would be useful to add bar plots including all 27 modules' p-values (maybe ordered by the p-values?) as supplementary figures.

Reply: As suggested, we have added Supplementary Figure S3 (below) to show all 27 modules and their correlation with BP:

Supplementary Figure S3. Co-expression modules and their correlation with BP. A) The correlation of eigengenes of each BP coexpression module (coEM) with SBP; B) The correlation of

eigengenes of each BP coEM with DBP; C) The enrichment of BP top signature genes in each BP coEM. The y-axis is the $-\log_{10}$ transformed p value.

1.2 Fisher's exact test is typically used for categorical data. Therefore, when performing SSEA, a GWAS p-value threshold must have been applied, but has not been specified. What is the threshold?

Reply: In this study, we used a GWAS p-value threshold of $p < 0.05$ for the Fisher's exact test in SSEA. We have added this information in the methods section of the revised manuscript (page 22): "In order to perform Fisher's exact test, we categorized all eSNPs into significant and non-significant categories based on their association with BP using a BP GWAS p-value threshold of $p < 0.05$."

1.3 It seems that Mann-Whitney/Wilcoxon ranked-sum test is more appropriate for SSEA, because the rank-sum test is mostly sensitive to changes in the median while KS-test is sensitive to ANY differences in the two distributions (i.e. shape, spread, or median). Therefore, I suggest that the authors also calculate the rank-sum test p-values.

Reply: We chose to use KS and Fisher's exact tests to specifically test whether there is significant deviation (toward lower p values) in the disease association p values of the eSNPs of BP-associated gene sets from the expected random distribution. Therefore, our intention was to test for differences in the distribution, not the mean or median p values. This approach has been used consistently in all our previous studies utilizing SSEA (Chan et al, 2014; Huan et al, 2013; Mäkinen et al, 2014). We recognize that testing the median differences using the Mann-Whitney/Wilcoxon test will also be supportive of our findings. As suggested by the reviewer, we used the Mann-Whitney/Wilcoxon ranked-sum (Wilcox) test to repeat our SSEA analysis. As shown in the Response Table 1 below, the new results using the Wilcox test are highly consistent with those from the KS and Fisher's exact tests, further confirming our results. However, we feel that reporting results from three separate tests may be confusing, especially with regard to small differences in the results from different tests. We therefore opted to keep our original results in the revised manuscript.

MODULE	SBP GWAS			DBP GWAS		
	KS Test Pval	Fisher Test Pval	Wilcox p val	KS Test Pval	Fisher Test Pval	Wilcox p val
BP signature	0.98	1	1	0.29	1	1
Turquoise	2.8e-45	8.0e-115	5.2e-52	1.8e-28	3.0e-39	9.5e-24
Blue	1.4e-44	7.0e-54	1.9e-34	1.3e-8	3.4e-15	9.8e-6
Red	8.0e-5	1.65E-17	7.8e-3	2.2e-15	6.7e-19	1.0e-13
Purple	0.65	0.58	1	1	1	1
Lightyellow	1.6e-3	1	0.96	0.12	1	1

Chocolate	2.2e-14	5.0e-5	1.3e-12	0.07	1	1
-----------	---------	--------	---------	------	---	---

Response Table 1: SNP set enrichment analysis of BP coexpression modules and BP signature gene set

1.4 To further strengthen the statistical significance of SSEA, the observed p-values should be compared to the distribution of p-values of random gene sets that match the number of genes.

Reply: We thank the reviewer for this suggestion. In the revised manuscript, we carried out 1000 permutations for each gene set to compare the observed p-values with those of random gene sets (matched for the number of genes). Permutation-based p values were derived using the formula (number of p values from random gene sets that are smaller than the observed p value/1000). The new results are shown in Table 3 on page 41 of the revised manuscript (also shown below). We are glad to report that the results from the suggested permutation analysis are highly consistent with our original results. The Turquoise, Blue, Red and Chocolate coEMs remained significant in the permutation-based analysis.

MODULE	SBP GWAS				DBP GWAS			
	KS P	Permutation-based KS P*	Fisher P	Permutation-based Fisher P*	KS P	Permutation-based KS P*	Fisher P*	Permutation-based Fisher P*
BP signature	0.98	0.96	1	1	0.20	0.23	1	1
Turquoise	2.8e-45	<0.001	7.8e-115	<0.001	1.8e-28	<0.001	3.0e-39	<0.001
Blue	1.4e-44	<0.001	7.0e-54	<0.001	1.3e-8	<0.001	3.4e-15	<0.001
Red	8.0e-5	<0.001	1.7e-17	<0.001	2.2e-15	<0.001	6.7e-19	<0.001
Purple	0.65	0.71	0.58	0.61	1	1	1	1
Lightyellow	1.6e-3	0.004	1	1	0.12	0.16	1	1
Chocolate	2.3e-14	<0.001	5.0e-5	<0.001	0.07	0.06	1	1

Main Text Table 3: SNP set enrichment analysis of BP coexpression modules and BP signature gene set

Highlighted p values pass Bonferroni-corrected $p < 0.05$;

**Permutation-based p is empirically derived based on 1000 permutations (see Methods). <0.001 indicates none of the 1000 random gene sets of matching size had p values lower than the observed test p values.*

In the revised manuscript, we revised the SSEA methods on page 22:

“... For each gene set and each statistical test (KS or Fisher’s exact test), we computed empirically derived enrichment p values based on 1000 permutations. Each permutation involved random sampling of equal number of genes matching the gene set being tested. Each permutation gene set was subject to the same KS or Fisher’s exact test as was used for the testing gene set. The empirically derived p value was estimated as the number of permutation gene sets with p values less than the observed p value of a given gene set/1000.”

1.5 In Table 5, how many of these top 20 KDs are also detected by the BioGrid PPI database?

Reply: Among the top 20 KDs, 13 were detected as KDs using the HPRD PPI database and the other 7 were from Bayesian networks. Six of the 13 (46%) HPRD PPI KDs were replicated using the BioGrid PPI database. Considering that the direct PPI overlap between the two databases is only 15% (as shown in our previous response letter), a replication rate of 46% for the top KDs supports stronger reliability of the top KDs. Since our comparison analysis between PPI databases revealed better performance of the HPRD PPI database (as reported in our previous response letter), we kept the top KDs from HPRD PPI in Table 5.

Reviewer #2:

The only remaining concern I have is that I do not agree with the authors that the 32 CoEMs shall not be corrected for multiple testing while correlating the eigengene values with hypertension (not necessarily bonferroni ($P_{\text{values}}/32$) but FDR-adjusted). I think this will become practice for manuscript in which associations are sought for network modules with phenotypes. I agree, however, that the downstream validation now is so convincing that there are little doubt, if any, of the true value of the HT causal co_EMs particular for the co-EM subnetwork surrounding SH2B3.

Reply: In the revised manuscript, we used the Benjamini-Hochberg method to correct for multiple tests and estimated the FDR. All 6 significant modules (at $p < 0.05$) passed $FDR < 0.2$, and the Chocolate module passed $FDR < 0.05$. We included this information in the revised manuscript on page 7:

“We identified 27 coEMs (Fig 2A and Supplementary Fig S2; the names of the coEMs are represented by different colors). Six coEMs (Turquoise, Blue, Red, Purple, Lightyellow, and Chocolate modules) were associated with either SBP or DBP at $p < 0.05$ and passed $FDR < 0.2$ (Fig 2B). The Chocolate module passed $FDR < 0.05$.”

Although $p < 0.05$ or $FDR < 0.2$ is arguably not a stringent cutoff, several previous coexpression network studies have demonstrated that interesting findings can be discovered when a less rigorous

statistical threshold is used to identify trait-associated coEMs (i.e. not corrected for multiple testing) (Farber, 2010; Leduc et al, 2012; Miller et al, 2008; Miller et al, 2013; Vanderlinden et al, 2013). An additional rationale for using nominal p values to select coEMs is that these coEMs are further integrated with multiple levels of additional information including genetic signals, eQTLs, and Bayesian and PPI gene networks. Integration of multiple levels of data has been proven to reduce false discoveries. Therefore, although the initial selection of coEMs did not involve extremely stringent cutoffs and was more inclusive, the additional downstream analyses serve to increase confidence in the findings. Last but not least, we were able to experimentally validate SH2B3 as a key driver of the Turquoise coEM, which only passed $p < 0.05$ and $FDR < 0.2$. This experimental proof further supports the notion that coEMs that show only nominal association with a trait can still be biologically valid and meaningful.

Reviewer #3:

3.1. Multiple-hypothesis correction was not conducted in many places in the text. For example, the authors identified 27 coEMs, among which 6 were associated with either SBP or DBP and 2 were associated with BP signature gene set. Since the enrichment test was performed across 27 coEMs, the uncorrected P values should be adjusted. In the same line, almost all the GO enrichment P values in this study were uncorrected. This also includes the enrichment test of KD genes.

Reply: As detailed in our response to Reviewer 2 comment, we have added the Benjamini-Hochberg method to correct for multiple testing and estimated FDR for the selection of BP coEMs. For the GO enrichment analysis, we previously provided all GO biological process terms meeting nominal $p < 0.01$ and marked those passing Bonferroni correction ($0.05/825$ unique GO biological process terms is $p < 6e-5$) in Table 2. We only focused on reporting the GO terms passing Bonferroni correction in the main text of the Results and Discussion sections. In the revised manuscript, we deleted GO terms that did not pass Bonferroni correction in Table 2 and changed all p values to Bonferroni-corrected p values.

For the Key Driver analysis, as described in the Methods section, the reported KD enrichment p values were already corrected by the number of genes (i.e., all potential KD candidates) in the gene network. Therefore, all KDs reported have passed Bonferroni-corrected $p < 0.05$. We have clarified this in the Methods section (page 23) of the revised manuscript as follows:

“A G (gene) that reached a Bonferroni-corrected KD-enrichment $p < 0.05$ was reported as a KD (after correction for the number of genes in the 3rd-layer expanding network of the tested BP causal gene set).”

3.2. SSEA analysis section: the size of each co-expression module should be considered. For larger modules with more genes, they may have higher chance to contain expression-SNPs/eQTLs or significance GWAS variants. Therefore, the authors need to control for module sizes in their statistical test.

Reply: As detailed in our response to Reviewer 1 (Comment 1.4), in the revised manuscript, we have added permutation-based analyses that control for module size. The new results are shown in Table 3 (page 41) of the revised manuscript (also shown below). We are glad to report that the results from permutation testing are highly consistent with our original results. The Turquoise, Blue, Red and Chocolate coEMs remained significant after controlling for module size.

Main Text Table 3: SNP set enrichment analysis of BP coexpression modules and BP signature gene set

MODULE	SBP GWAS				DBP GWAS			
	KS_P	Permutatio n-based KS P*	Fisher P	Permutatio n-based Fisher P *	KS_P	Permutatio n-based KS P*	Fisher P*	Permutatio n-based Fisher P*
BP signature	0.98	0.96	1	1	0.20	0.23	1	1
Turquoise	2.8e-45	<0.001	7.8e-115	<0.001	1.8e-28	<0.001	3.0e-39	<0.001
Blue	1.4e-44	<0.001	7.0e-54	<0.001	1.3e-8	<0.001	3.4e-15	<0.001
Red	8.0e-5	<0.001	1.7e-17	<0.001	2.2e-15	<0.001	6.7e-19	<0.001
Purple	0.65	0.71	0.58	0.61	1	1	1	1
Lightyellow	1.6e-3	0.004	1	1	0.12	0.16	1	1
Chocolate	2.3e-14	<0.001	5.0e-5	<0.001	0.07	0.06	1	1

Highlighted p values pass Bonferroni-corrected $p < 0.05$;

*Permutation-based p is empirically derived based on 1000 permutations (see Methods). <0.001 indicates none of the 1000 random gene sets of matching size had p values lower than the observed test p values.

3.3 The use of "putative causal" throughout the manuscript is too strong. The enrichment at most only suggests association, but not putative causal.

Reply: We agree with the reviewer that any strong claims of causality are unfounded without experimental validation. We and others have used the phrase “*putatively causal*” to refer to gene sets that are inferred to play a putative causal role in diseases based on genetic evidence. Our enrichment analysis considered genetic association, and genetic variation is upstream of disease events based on the central dogma. This concept forms the basis of Mendelian randomization analysis, which helps to tease out causal mechanisms of disease. Therefore, the results from our SSEA enrichment analysis do not simply imply association, but carry “putatively” causal information. Our validation experiments in the SH2B3 knockout mouse further support the “putatively” causal nature of coEMs and their key drivers. Taken together, we feel that it is important to reflect the nature of the SSEA analysis that is designed to infer potential causality. Based on the reviewer’s objections to the word “putative”, we have reworded “putative causal” to “genetically inferred causal” throughout the manuscript to indicate they are potentially causal based on statistical inference, but the causal nature is in need of future experimental validation. In the revised manuscript (page 9), we clarified this point as follows:

“A BP gene set showing significance in SSEA is referred to as “genetically inferred causal”, because it is supported by orthogonal genetic evidence (i.e. association of its eSNPs with BP in GWAS) that is unlikely to be confounded by non-genetic factors. The same term also implies that further experimental validation is needed to establish causality with certainty.”

3.4. The definition of KD genes is unclear. From what described in the text, A KD gene is defined (i) a local network hub (ii) its network neighborhood shows BP gene enrichment. It is thus unclear if a KD gene itself is associated with blood pressure or shows differential expression in patients. From the definition given, KD genes could have neither, which is more evidenced against than support their roles as "driver". More importantly, for many genes causal for human diseases, they do not have to be hubs with global impacts on the network; instead, disease might arise by simply ablating the peripheral nodes with a highly specified function. Therefore, the authors need to carefully justify why the "driver" genes in BP traits should be network hubs.

Reply: We defined KDs as key regulatory genes of disease-related networks based on network topology, but not from disease association information of the KDs themselves. As pointed out by the reviewer, a KD is a hub of a gene network that is enriched for BP genes. Due to the “hub” properties of the KDs, perturbations in KDs are more likely to modulate a large number of BP-associated genes in a BP-related gene subnetwork, which in turn affect BP. Therefore, these genes possess properties of key regulators or drivers of BP. In this context, “driver” genes and “causal” genes are two separate concepts: causal genes can be either hub or peripheral nodes as long as their perturbations lead to disease phenotypes regardless of the magnitude of effect, whereas driver genes can only be hub nodes whose perturbations, if induced, may cause strong phenotypic effects due to their

potential to regulate many disease associated genes. The importance of KD genes has been supported by a number of recent high impact studies (Chan et al, 2014; Goh et al, 2007; Mäkinen et al, 2014; Wang et al, 2012; Zhang et al, 2013).

3.5. In the regulatory association network with SH2B3, the authors showed that many known BP-related genes were involved. Are there any enrichment statistics to support this observation?

Reply: In the revised manuscript, we systematically checked the overlap of known BP-related genes with genes in the *SH2B3* derived PPI network. We used the latest version of GeneRif (<http://www.ncbi.nlm.nih.gov/gene/about-generif>; downloaded in Jan 2015) and searched for BP-related genes using keywords “hypertension” and “blood pressure”. We found 657 genes as suggestive BP-related genes in GeneRif (GeneRif collected literature descriptions of 14,069 unique human genes). Among the 657 BP-related genes, 41 were present in the *SH2B3* derived PPI subnetwork comprised of 362 genes. Comparison of the two ratios 656/14,069 and 41/362 yielded $p=5.5e-8$ (by the hypergeometric test) and 2.43-fold enrichment. This result indicates that the *SH2B3* derived PPI subnetwork is enriched for known BP-related genes (based on literature support). We have included this information in the revised manuscript (page 12).

*“In order to systematically check if the SH2B3 derived PPI subnetwork showed any enrichment for literature-based BP-related genes, we created a list of 657 BP-related genes by searching GeneRif (<http://www.ncbi.nlm.nih.gov/gene/about-generif>; downloaded in Jan 2015) using the keywords “hypertension” and “blood pressure”. GeneRif includes literature descriptions of 14,069 unique human genes in total. We found that 41 of the 657 genes were present in the SH2B3 derived PPI subnetwork, which consisted of 362 genes in total, including *PLCE1* (Ehret et al, 2011), *BAT2* (Ehret et al, 2011), *ADRB2* (Lou et al, 2010), *RHOA* (Connolly & Aaronson, 2011), and *SOCS1* (Satou et al, 2012). Comparison of the two ratios 656/14,069 and 41/362 yielded $p=5.5e-8$ (by the hypergeometric test) and 2.43-fold enrichment. This result indicates that the SH2B3 derived PPI subnetwork is enriched for known BP-related genes.”*

3.6. The overall co-expression network was constructed using the entire selected participant in this study, 28% (11%+17%) with hypertension or pre-hypertension. The author should examine whether the sample heterogeneity would affect their conclusions. A recent study using co-expression network to study autism (Voineagu, I. et al. 2011, Nature) constructed the networks separately for cases and controls, and observed modules showing differences between the two groups. The authors might try to divide the samples according to their BP traits and examine the module differences between individuals with and without hypertension traits, which is complementary to the differential expression study. Given almost ~4000 samples were studied in this study, the sub-division will unlikely compromise the statistical power in the comparisons.

Reply: The method suggested by reviewer is suitable for categorical or dichotomized traits (i.e., cases vs controls). As the reviewer pointed out, the study of autism (Voineagu, I. et al. 2011, Nature) constructed and compared coexpression networks between cases and controls, and identified discrete coexpression modules associated with autism. We have also recently utilized a similar approach to identify the disrupted coexpression network modules in 188 coronary heart disease case-control pairs (Huan et al, 2013).

In this study, however, the traits of interest are continuous blood pressure traits (systolic and diastolic BP) that follow a normal distribution. The most suitable coexpression network methodology for continuous traits is the type of analysis we have implemented in the current study. The same coexpression network approach has been employed by the developers of coexpression network methodology when studying continuous traits such as mouse weight (Ghazalpour et al, 2006) and age (Horvath et al, 2012).

The suggestion that we compare coexpression networks in hypertensive cases vs. normotensive controls might be of interest, but transforming quantitative/continuous traits into categorical/dichotomized traits will result in a loss of statistical power and will be a very different design that is beyond the scope of this study.

3.7. In many places, the manuscript cited unpublished results and conclusions. Is it a companion paper sent to MSB, or the authors might consider using other alternative sources to support their data.

Reply: Our related study, which shows that *SH2B3* (also known as *LNK*) plays a key role in the development of hypertension in mice, has been recently accepted by *The Journal of Clinical Investigation* (Saleh et al, 2015). We cited this paper in the revised manuscript to support our results.

3.8. 83 genes showed differential expression between cases and controls, among which 65 and 8 showed positive and negative correlations with BP traits, respectively. The authors should provide the related statistics (distribution of correlations) in the context and supplementary tables to help the audience better understand the relative contribution of each gene to the BP traits.

Reply: In Supplementary Data 1 and 2, we included beta coefficients (estimates) of differentially expressed BP genes, with a positive value indicating a positive correlation, and a negative value indicating a negative correlation.

3.9. Overall, this paper shows much "technical development" to uncover the loci or gene groups associated with BP traits, but their potential biology remains less discussed. It seems that the authors did not give much attention to the 83 differentially expressed genes, and relied more on the GWAS signals for eSNPs. It will be useful to discuss their disease implications.

Reply: We have expanded the discussion of the top signature genes in the revised manuscript on (page 14) as follows:

*"By first applying a traditional approach that focused on differentially expressed individual genes, we identified a gene signature set comprised of 83 genes whose expression levels were correlated with BP traits. The 83-gene BP signature gene set did not show significant enrichment for biological processes or pathways suggesting that the traditional single-gene approach lacks power to capture high-order organization of genes underlying BP regulation. Subsequent SNP set enrichment analysis (SSEA), which incorporates genetic signals, did not support an overall causal role of the top BP signature genes. Although this lack of overall significance in SSEA does not exclude a small subset of genes being causal -- for example, a top signature gene *ATP2B1* has been previously detected as a GWAS signal in ICBP GWAS at $p < 5 \times 10^{-8}$. *ATP2B1* (*ATPase, Ca²⁺ transporting, plasma membrane 1*) is known to be responsible for *Ca²⁺ transportation in plasma membrane* and a BP-associated *ATP2B1* SNP has been linked to *ATP2B1* expression in umbilical artery smooth muscle cells (Tabara et al, 2010). *Ca²⁺* is critical for muscle contraction (Marks, 2003) and defects or altered expression of *ATP2A1* will likely induce changes in artery smooth muscle contraction which may in turn affects blood pressure variability. Another top signature gene, *FOS* (known as *c-fos*), has been found to be associated with hypertension (Cunningham et al, 2006; Minson et al, 1996). The *c-fos* gene is considered to be a useful marker of neuronal activity in different sites, including those important in BP control. In the rat, *c-fos* expression in the brain is likely to be important for BP control; and the blockade of *c-fos* expression in this region attenuates resting and stimulated BP levels. Inhibition of local neuronal activity acutely increased both BP and immunoreactivity to *Fos*, the protein product of the *c-fos* gene. Intravenous infusion of sodium nitroprusside induced hypotension and the number of *Fos*-positive spinal sympathetic neurons increased (Minson et al, 1996). Several additional BP signature genes have been reported to be involved BP-related diseases or processes such as cardiovascular disease (e.g., *ABCA1* (Tang & Oram, 2009), *AHR*(Zhang, 2011), and *GZMB* (Joehanes et al, 2013)), type II diabetes (e.g., *ABCA1* (Tang & Oram, 2009), *ANXA1* (Lindgren et al, 2001), and *PTGS2* (Shanmugam et al, 2006)), and inflammation (e.g., *GZMB* (Hiebert & Granville, 2012) and *KLRD1* (Choi et al, 2012)). We speculate that these genes may play important roles in BP regulation, but further mechanistic studies are necessary."*

References:

Benjamini Y, Hochberg Y (1995) Controlling the false discovery rate: a practical and powerful approach to multiple testing. *Journal of the Royal Statistical Society Series B (Methodological)*: 289-300

Chan KHK, Huang Y-T, Meng Q, Wu C, Reiner A, Sobel EM, Tinker L, Lusis AJ, Yang X, Liu S (2014) Shared Molecular Pathways and Gene Networks for Cardiovascular Disease and Type 2 Diabetes Mellitus in Women Across Diverse Ethnicities. *Circulation: Cardiovascular Genetics* 7: 911-919

- Choi HJ, Yun HS, Kang HJ, Ban H-J, Kim Y, Nam H-Y, Hong E-J, Jung S-Y, Jung SE, Jeon J-P (2012) Human transcriptome analysis of acute responses to glucose ingestion reveals the role of leukocytes in hyperglycemia-induced inflammation. *Physiological genomics* **44**: 1179-1187
- Connolly MJ, Aaronson PI (2011) Key role of the RhoA/Rho kinase system in pulmonary hypertension. *Pulmonary pharmacology & therapeutics* **24**: 1-14
- Cunningham JT, Fleming T, Penny ML, Herrera-Rosales M, Mifflin SW (2006) Increased c-Fos in medullary cardiovascular nuclei in acute and chronic renal wrap hypertension. *The FASEB Journal* **20**: A1205-A1206
- Ehret GB, Munroe PB, Rice KM, Bochud M, Johnson AD, Chasman DI, Smith AV, Tobin MD, Verwoert GC, Hwang SJ, Pihur V, Vollenweider P, O'Reilly PF, Amin N, Bragg-Gresham JL, Teumer A, Glazer NL, Launer L, Zhao JH, Aulchenko Y et al (2011) Genetic variants in novel pathways influence blood pressure and cardiovascular disease risk. *Nature* **478**: 103-109
- Farber CR (2010) Identification of a gene module associated with BMD through the integration of network analysis and genome-wide association data. *Journal of Bone and Mineral Research* **25**: 2359-2367
- Ghazalpour A, Doss S, Zhang B, Wang S, Plaisier C, Castellanos R, Brozell A, Schadt EE, Drake TA, Lusis AJ, Horvath S (2006) Integrating genetic and network analysis to characterize genes related to mouse weight. *PLoS Genet* **2**: e130
- Goh K-I, Cusick ME, Valle D, Childs B, Vidal M, Barabási A-L (2007) The human disease network. *Proceedings of the National Academy of Sciences* **104**: 8685-8690
- Hiebert PR, Granville DJ (2012) Granzyme B in injury, inflammation, and repair. *Trends in molecular medicine* **18**: 732-741
- Horvath S, Zhang Y, Langfelder P, Kahn RS, Boks MP, van Eijk K, van den Berg LH, Ophoff RA (2012) Aging effects on DNA methylation modules in human brain and blood tissue. *Genome Biol* **13**: R97
- Huan T, Zhang B, Wang Z, Joehanes R, Zhu J, Johnson AD, Ying S, Munson PJ, Raghavachari N, Wang R, Liu P, Courchesne P, Hwang SJ, Assimes TL, McPherson R, Samani NJ, Schunkert H, Meng Q, Suver C, O'Donnell CJ et al (2013) A systems biology framework identifies molecular underpinnings of coronary heart disease. *Arterioscler Thromb Vasc Biol* **33**: 1427-1434
- Joehanes R, Ying S, Huan T, Johnson AD, Raghavachari N, Wang R, Liu P, Woodhouse KA, Sen SK, Tanriverdi K (2013) Gene expression signatures of coronary heart disease. *Arteriosclerosis, thrombosis, and vascular biology* **33**: 1418-1426
- Leduc MS, Blair RH, Verdugo RA, Tsaih S-W, Walsh K, Churchill GA, Paigen B (2012) Using bioinformatics and systems genetics to dissect HDL cholesterol levels in an MRL/MpJ x SM/J intercross. *Journal of lipid research: jlr*. M025833
- Levy D, Ehret GB, Rice K, Verwoert GC, Launer LJ, Dehghan A, Glazer NL, Morrison AC, Johnson AD, Aspelund T, Aulchenko Y, Lumley T, Kottgen A, Vasun RS, Rivadeneira F, Eiriksdottir G, Guo X, Arking DE, Mitchell GF, Mattace-Raso FU et al (2009) Genome-wide association study of blood pressure and hypertension. *Nat Genet* **41**: 677-687
- Li Y, Willer CJ, Ding J, Scheet P, Abecasis GR (2010) MaCH: using sequence and genotype data to estimate haplotypes and unobserved genotypes. *Genetic epidemiology* **34**: 816-834
- Lindgren CM, Nilsson A, Orho-Melander M, Almgren P, Groop LC (2001) Characterization of the annexin I gene and evaluation of its role in type 2 diabetes. *Diabetes* **50**: 2402-2405

- Lou Y, Liu J, Huang Y, Liu J, Wang Z, Liu Y, Li Z, Li Y, Xie Y, Wen S (2010) A46G and C79G polymorphisms in the β 2-adrenergic receptor gene (ADRB2) and essential hypertension risk: a meta-analysis. *Hypertension Research* **33**: 1114-1123
- Mäkinen V-P, Civelek M, Meng Q, Zhang B, Zhu J, Levian C, Huan T, Segrè AV, Ghosh S, Vivar J (2014) Integrative genomics reveals novel molecular pathways and gene networks for coronary artery disease. *PLoS genetics* **10**: e1004502
- Marks AR (2003) Calcium and the heart: a question of life and death. *Journal of Clinical Investigation* **111**: 597
- Miller JA, Oldham MC, Geschwind DH (2008) A systems level analysis of transcriptional changes in Alzheimer's disease and normal aging. *The Journal of Neuroscience* **28**: 1410-1420
- Miller JA, Woltjer RL, Goodenbour JM, Horvath S, Geschwind DH (2013) Genes and pathways underlying regional and cell type changes in Alzheimer's disease. *Genome Med* **5**: 48
- Minson J, Arnolda L, Llewellyn-Smith I, Pilowsky P, Chalmers J (1996) Altered c-fos in rostral medulla and spinal cord of spontaneously hypertensive rats. *Hypertension* **27**: 433-441
- Saleh MA, McMaster WG, Wu J, Norlander AE, Funt SA, Thabet SR, Kirabo A, Xiao L, Chen W, Itani HA, Michell D, Huan T, Zhang Y, Titze J, Levy D, Harrison DG, Madhur MS (2015) Lymphocyte adaptor protein LNK deficiency exacerbates hypertension and end-organ inflammation. *The Journal of Clinical Investigation* **In press**
- Satou R, Miyata K, Gonzalez-Villalobos RA, Ingelfinger JR, Navar LG, Kobori H (2012) Interferon- γ biphasically regulates angiotensinogen expression via a JAK-STAT pathway and suppressor of cytokine signaling 1 (SOCS1) in renal proximal tubular cells. *The FASEB Journal* **26**: 1821-1830
- Shanmugam N, Todorov I, Nair I, Omori K, Reddy M, Natarajan R (2006) Increased expression of cyclooxygenase-2 in human pancreatic islets treated with high glucose or ligands of the advanced glycation endproduct-specific receptor (AGER), and in islets from diabetic mice. *Diabetologia* **49**: 100-107
- Tabara Y, Kohara K, Kita Y, Hirawa N, Katsuya T, Ohkubo T, Hiura Y, Tajima A, Morisaki T, Miyata T (2010) Common Variants in the ATP2B1 Gene Are Associated With Susceptibility to Hypertension The Japanese Millennium Genome Project. *Hypertension* **56**: 973-980
- Takaki S, Sauer K, Iritani BM, Chien S, Ebihara Y, Tsuji K, Takatsu K, Perlmutter RM (2000) Control of B cell production by the adaptor protein lnk. Definition Of a conserved family of signal-modulating proteins. *Immunity* **13**: 599-609
- Tang C, Oram JF (2009) The cell cholesterol exporter ABCA1 as a protector from cardiovascular disease and diabetes. *Biochimica et Biophysica Acta (BBA)-Molecular and Cell Biology of Lipids* **1791**: 563-572
- Thorvaldsdóttir H, Robinson JT, Mesirov JP (2012) Integrative Genomics Viewer (IGV): high-performance genomics data visualization and exploration. *Briefings in bioinformatics*: bbs017
- Vanderlinden LA, Saba LM, Kechris K, Miles MF, Hoffman PL, Tabakoff B (2013) Whole brain and brain regional coexpression network interactions associated with predisposition to alcohol consumption. *PLoS One* **8**: e68878
- Wang I, Zhang B, Yang X, Zhu J, Stepaniants S, Zhang C, Meng Q, Peters M, He Y, Ni C (2012) Systems analysis of eleven rodent disease models reveals an inflammatome signature and key drivers. *Molecular systems biology* **8**

Zhang B, Gaiteri C, Bodea L-G, Wang Z, McElwee J, Podtelezchnikov AA, Zhang C, Xie T, Tran L, Dobrin R (2013) Integrated systems approach identifies genetic nodes and networks in late-onset Alzheimer's disease. *Cell* **153**: 707-720

Zhang N (2011) The role of endogenous aryl hydrocarbon receptor signaling in cardiovascular physiology. *Journal of cardiovascular disease research* **2**: 91-95

3rd Editorial Decision

02 March 2015

Thank you for sending us your revised manuscript. I apologize for the somewhat delayed response, which was due to the fact that last week I was out of the office for a few days, performing lab visits. We are now satisfied with the modifications made and we think that the study is suitable for publication.

Before we formally accept the manuscript we would like to ask you to provide some further information regarding the newly identified eSNPs (the full list of which will be published in Joehanes et al., in preparation) in order to facilitate the use of this new data by those interested, until the related paper is accepted for publication. In particular, we would ask you to mark in Supplementary Dataset S3 the newly identified eSNPs/eQTLs (i.e. by an asterisk) and to provide information i.e. on the allele that is affected and how it is linked to the expression of the related gene.

3rd Revision - authors' response

08 March 2015

Thank you for the encouraging decision! We have revised our manuscript in accordance with your suggestions as detailed below.

Comment #1: Before we formally accept the manuscript we would like to ask you to provide some further information regarding the newly identified eSNPs (the full list of which will be published in Joehanes et al., in preparation) in order to facilitate the use of this new data by those interested, until the related paper is accepted for publication. In particular, we would ask you to mark in Supplementary Dataset S3 the newly identified eSNPs/eQTLs (i.e. by an asterisk) and to provide information i.e. on the allele that is affected and how it is linked to the expression of the related gene.

Response: We have included the details of newly identified eSNPs (Joehanes et al., in preparation) in Supplementary Dataset S3 but specifying the eSNP sources. If the eSNPs are from the unpublished FHS study, we added columns detailing the allele affected and the direction of association with the expression of the related gene.